# Joints in Random Forests

**Alvaro H. C. Correia**
a.h.chaim.correia@tue.nl
Eindhoven University of Technology

**Robert Peharz**
r.peharz@tue.nl
Eindhoven University of Technology

**Cassio de Campos**
c.decampos@tue.nl
Eindhoven University of Technology

## Abstract

Decision Trees (DTs) and Random Forests (RFs) are powerful discriminative learners and tools of central importance to the everyday machine learning practitioner and data scientist. Due to their discriminative nature, however, they lack principled methods to process inputs with missing features or to detect outliers, which requires pairing them with imputation techniques or a separate generative model. In this paper, we demonstrate that DTs and RFs can naturally be interpreted as generative models, by drawing a connection to Probabilistic Circuits, a prominent class of tractable probabilistic models. This reinterpretation equips them with a full joint distribution over the feature space and leads to Generative Decision Trees (GeDTs) and Generative Forests (GeFs), a family of novel hybrid generative-discriminative models. This family of models retains the overall characteristics of DTs and RFs while additionally being able to handle missing features by means of marginalisation. Under certain assumptions, frequently made for Bayes consistency results, we show that consistency in GeDTs and GeFs extend to any pattern of missing input features, if missing at random. Empirically, we show that our models often outperform common routines to treat missing data, such as K-nearest neighbour imputation, and moreover, that our models can naturally detect outliers by monitoring the marginal probability of input features.

## 1 Introduction

Decision Trees (DTs) and Random Forests (RFs) are probably the most widely used non-linear machine learning models of today. While Deep Neural Networks are in the lead for image, video, audio, and text data—likely due to their beneficial inductive bias for signal-like data—DTs and RFs are, by and large, the default predictive model for tabular, domain-agnostic datasets. Indeed, Kaggle's 2019 report on the *State of Data Science and Machine Learning* [20] lists DTs and RFs as second most widely used techniques, right after linear and logistic regressions. Moreover, a study by Fernandez et al. [12] found that RFs performed best on 121 UCI datasets against 179 other classifiers. Thus, it is clear that DTs and RFs are of central importance for the current machine learning practitioner.

DTs and RFs are generally understood as *discriminative models*, that is, they are solely interpreted as predictive models, such as *classifiers* or *regression functions*, while attempts to additionally interpret them as *generative* models are scarce. In a nutshell, the difference between discriminative and generative models is that the former aim to capture the *conditional distribution* $P(Y \mid \mathbf{X})$, while the latter aim to capture the whole *joint distribution* $P(Y, \mathbf{X})$, where $\mathbf{X}$ are the input features and $Y$ is the variable to be predicted—discrete for classification and continuous for regression. In this paper, we focus on classification, but the extension to regression is straightforward.

Generative and discriminative models are rather complementary in their strengths and use cases. While discriminative models typically fare better in predictive performance, generative models allow to analyse and capture the structure present in the input space. They are also "all-round predictors", that is, not restricted to a single prediction task but also capable of predicting any $X$ given $Y \cup \mathbf{X} \setminus X$. Moreover, generative models have some crucial advantages on the prediction task $P(Y \mid \mathbf{X})$ a discriminative model has been trained on, as they naturally allow to *detect outliers* (by monitoring $P(\mathbf{X})$) and *treat missing features* (by marginalisation). A purely discriminative model does not have any "innate" mechanisms to deal with these problems, and needs to be supported with a generative model $P(\mathbf{X})$ (to detect outliers) or imputation techniques (to handle missing features).

Ideally, we would like the best of both worlds: having the good predictive performance of discriminative models *and* the advantages of generative models. In this paper, we show that this is achievable for DTs and RFs by relating them to *Probabilistic Circuits* (PCs) [50], a class of generative models based on computational graphs of *sum nodes* (mixtures), *product nodes* (factorisations), and *leaf nodes* (distribution functions). PCs subsume and represent a wide family of related models, such as *arithmetic circuits* [8], *AND/OR-graphs* [31], *sum-product networks* [38], *cutset networks* (CNets) [44], and *probabilistic sentential decision diagrams* [24]. While many researchers are aware of the similarity between DTs and PCs—most notably, CNets [44] can be seen as a type of generative DT—the connection to classical, discriminative DTs [40] and RFs [3] has not been studied so far.

We show that DTs and RFs can be naturally cast into the PC framework. For any given DT, we can construct a corresponding PC, a *Generative Decision Tree* (GeDT), representing a full joint distribution $P(Y, \mathbf{X})$. This distribution gives rise to the predictor $P(Y \mid \mathbf{X}) = {P(Y, \mathbf{X})} / {\sum_y P(y, \mathbf{x})}$, which is identical to the original DT, if we impose certain constraints on the conversion from DT to GeDT. Additionally, a GeDT also fits the joint distribution $P(\mathbf{X})$ to the training data, "upgrading" the DT to a fully generative model. For a *completely observed sample* $\mathbf{X} = \mathbf{x}$, the original DT and a corresponding GeDT agree entirely (yield the exact same predictions), and moreover, have the *same computational complexity* (a discussion on time complexity is deferred to the supp. material). By converting each DT in an RF into an GeDT, we obtain an ensemble of GeDTs, which we call *Generative Forest* (GeF). Clearly, if each GeDT in a GeF agrees with its original DT, then GeFs also agree with their corresponding RFs.

GeDTs and GeFs have a crucial advantage in the case of *missing features*, that is, assignments $\mathbf{X}_o = \mathbf{x}_o$ for some subset $\mathbf{X}_o \subset \mathbf{X}$, while $\mathbf{X}_{\neg o} = \mathbf{X} \setminus \mathbf{X}_o$ are *missing at random*. In a GeDT, we can marginalise the missing features and yield the predictor

$$P(Y \mid \mathbf{X}_o) = \frac{\int_{\mathbf{x}_{\neg o}} P(Y, \mathbf{X}_o, \mathbf{x}_{\neg o}) \mathrm{d}\mathbf{x}_{\neg o}}{\sum_y \int_{\mathbf{x}_{\neg o}} P(y, \mathbf{X}_o, \mathbf{x}_{\neg o}) \mathrm{d}\mathbf{x}_{\neg o}}. \tag{1}$$

For GeFs, we yield a corresponding ensemble predictor for missing features, by applying marginalisation to each GeDT. Using the true data generating distribution in Eq. (1) would deliver the *Bayes optimal predictor* for any subset $\mathbf{X}_o$ of observed features. Thus, since GeDTs are trained to approximate the true distribution, using the predictor of Eq. (1) under missing data is well justified. We show GeDTs are in fact *consistent*: they converge to the Bayes optimal classifier as the number of data points goes to infinity. Our proof requires similar assumptions to those of previous results for DTs [1, 4, 17] but is substantially more general: while consistency in DTs is shown only for a classifier $P(Y \mid \mathbf{X})$ using fully observed samples, our consistency result holds for *all* $2^{|X|}$ classifiers $P(Y \mid \mathbf{X}_o)$: one for each observation pattern $\mathbf{X}_o \subseteq \mathbf{X}$. While the high-dimensional integrals in Eq. (1) seem prohibitive, they are in fact *tractable*, since a remarkable feature of PCs is that computing any marginal has *the same complexity* as evaluating the full joint, namely linear in the *circuit size*.

This ability of our models is desirable, as there is no clear consensus on how to deal with missing features in DTs at test time: The most common strategy is to use *imputation*, e.g. *mean* or *k-nearest-neighbour (KNN) imputation*, and subsequently feed the completed sample to the classifier. DTs also have two "built-in" methods to deal with missing features that do not require external models. These are the so-called *surrogate splits* [49] and an unnamed method proposed by Friedman in 1977 [14, 41]. Among these, KNN imputation seems to be the most widely used, and typically delivers good results on real-world data. However, we demonstrate it does not lead to a consistent predictor under missing data, even when assuming idealised settings. Moreover, in our experiments, we show that GeF classification under missing inputs often outperforms standard RFs with KNN imputation.

Our generative interpretation can be easily incorporated in existing DT learners and does not require drastic changes in the learning and application practice for DTs and RFs. Essentially, any DT algorithm can be used to learn GeDTs, requiring only minor bookkeeping and some extra generative learning steps. There are de facto no model restrictions concerning the additional generative learning steps, representing a generic scheme to augment DTs and RFs to generative models.

## 2   Notation and Background

In this paper we focus on classification tasks. To this end, let the set of explanatory variables (features) be $\mathbf{X} = \{X_1, X_2, \ldots, X_m\}$, where continuous $X_i$ assume values in some compact set $\mathcal{X}_i \subset \mathbb{R}$ and discrete $X_i$ assume values in $\mathcal{X}_i = \{1, \ldots, K_i\}$, where $K_i$ is the number of states for $X_i$. Let the joint *feature space* of $\mathbf{X}$ be denoted as $\boldsymbol{\mathcal{X}}$. We denote joint states, i.e. elements from $\boldsymbol{\mathcal{X}}$, as $\mathbf{x}$ and let $\mathbf{x}[i]$ be the state in $\mathbf{x}$ belonging to $X_i$. The class variable is denoted as $Y$, assuming values in $\mathcal{Y} = \{1, \ldots, K\}$, where $K$ is the number of classes. We assume that the pair $(\mathbf{X}, Y)$ is drawn from a fixed joint distribution $\mathbb{P}^*(\mathbf{X}, Y)$ which has density $p^*(\mathbf{X}, Y)$. While the true distribution $\mathbb{P}^*$ is unknown, we assume that we have a dataset $\mathcal{D}_n = \{(\mathbf{x}_1, y_1), \ldots, (\mathbf{x}_n, y_n)\}$ of $n$ i.i.d. samples from $\mathbb{P}^*$. When describing a directed graph $\mathcal{G}$, we refer to its set of nodes as $V$, reserving letters $u$ and $v$ for individual nodes. We denote the set of children and parents of a node $v$ as $\mathrm{ch}(v)$ and $\mathrm{pa}(v)$, respectively. Nodes $v$ without children are referred to as *leaves*, and nodes without parents are referred to as *roots*.

**Decision Trees.** A *decision tree* (DT) is based on a *rooted directed tree* $\mathcal{G}$, i.e. an acyclic directed graph with exactly one root $v_r$ and whose other nodes have exactly one parent. Each node $v$ in the DT is associated with a *cell* $\boldsymbol{\mathcal{X}}_v$, which is a subset of the feature space $\boldsymbol{\mathcal{X}}$. The cell of the root node $v_r$ is the whole $\boldsymbol{\mathcal{X}}$. The *child cells* of node $v$ form a partition of $\boldsymbol{\mathcal{X}}_v$, i.e. $\bigcup_{u \in \mathrm{ch}(v)} \boldsymbol{\mathcal{X}}_u = \boldsymbol{\mathcal{X}}_v$, $\boldsymbol{\mathcal{X}}_u \cap \boldsymbol{\mathcal{X}}_{u'} = \emptyset$, $\forall u, u' \in \mathrm{ch}(v)$. These partitions are usually defined via *axis-aligned splits*, by associating a decision variable $X_i$ to $v$, and partitioning the cell according to some rule on $X_i$'s values. Formally, we first project $\boldsymbol{\mathcal{X}}_v$ onto its $i^{\text{th}}$ coordinate, yielding $\mathcal{X}_{i,v} := \{\mathbf{x}[i] \mid \mathbf{x} \in \boldsymbol{\mathcal{X}}_v\}$, and construct a partition $\{\mathcal{X}_{i,u}\}_{u \in \mathrm{ch}(v)}$ of $\mathcal{X}_{i,v}$. The child cells are then given by $\boldsymbol{\mathcal{X}}_u = \{\mathbf{x} \mid \mathbf{x} \in \boldsymbol{\mathcal{X}}_v \wedge \mathbf{x}[i] \in \mathcal{X}_{i,u}\}$. Common choices for this partition are *full splits* for discrete variables, i.e. choosing $\{\mathcal{X}_{i,u}\}_{u \in \mathrm{ch}(v)} = \{\{x_i\}\}_{x_i \in \mathcal{X}_{i,v}}$ where children $u$ and states $x_i$ are in one-to-one correspondence, and *thresholding* for continuous variables, i.e. choosing $\{\mathcal{X}_{i,u}\}_{u \in \mathrm{ch}(v)} = \{\{x_i < t\}, \{x_i \geq t\}\}$ for some threshold $t$. Note that the leaf cells of a DT represent a partition $\boldsymbol{\mathcal{A}}$ of the feature space $\boldsymbol{\mathcal{X}}$. We denote the elements of $\boldsymbol{\mathcal{A}}$ as $\mathcal{A}$ and define $\mathcal{A}_v = \boldsymbol{\mathcal{X}}_v$ for each leaf $v$. A *DT classifier* is constructed by equipping each $\mathcal{A} \in \boldsymbol{\mathcal{A}}$ with a classifier $f^{\mathcal{A}} : \mathcal{A} \mapsto \Delta^K$, where $\Delta^K$ is the set of probability distributions over $K$ classes, i.e. $f^{\mathcal{A}}$ is a conditional distribution defined on $\mathcal{A}$. This distribution is typically stored as absolute class counts of the training samples contained in $\mathcal{A}$.

The overall DT classifier is given as $f(\mathbf{x}) = f^{\mathcal{A}(\mathbf{x})}(\mathbf{x})$ where $\mathcal{A}(\mathbf{x})$ is the leaf cell containing $\mathbf{x}$; $\mathcal{A}(\mathbf{x})$ is found by parsing the DT top-down, following the partitions (decisions) consistent with $\mathbf{x}$. This formulation captures the vast majority of DT classifiers proposed in the literature, notably CART [4] and ID3 [40]. The probably most widely used variant of DTs—which we also assume in this paper—is to define $f^{\mathcal{A}}$ as a constant function, returning the class proportions in cell $\mathcal{A}$. The $\arg\max$ of $f^{\mathcal{A}}(\mathbf{x})$ is equivalent to majority voting among all training samples which fall into the same cell. When learning a DT, the number of available training samples per cell reduces quickly, which leads to overfitting and justifies the need for pruning techniques [4, 32, 40, 42].

**Random Forests.** *Random Forests* (RFs) are ensembles of DTs which effectively counteract overfitting. Each DT in a RF is learned in a randomised fashion by, at each learning step, drawing a random sub-selection of variables containing only a fraction $p$ of all variables, where typical values are $p = 0.3$ or $p = \sqrt{m}$. The resulting DTs are not pruned but made "deep" until each leaf cell contains either only samples of one class or less than $T$ samples, where typical values are $T \in \{1, 5, 10\}$. This yields low bias, but high variance in the randomised DTs, which makes them good candidates for *bagging* (bootstrap aggregation) [19]. Thus, to further increase the variability among the trees, each of them is learned on a *bootstrapped* version of the training data [3].

**Probabilistic Circuits.** In this paper, we relate DTs to *Probabilistic Circuits* (PCs) [50], a family of density representations facilitating many exact and efficient inference routines. PCs are, like DTs, based on a rooted acyclic directed graph $\mathcal{G}$, albeit one with different semantics. PCs are computational graphs with three types of nodes, namely i) distribution nodes, ii) sum nodes and iii) product nodes.

**Input**  : Decision Tree $\mathcal{G}$ and training data $\mathcal{D}$
**Output** : Probabilistic Circuit $\mathcal{G}'$
let $\mathcal{G}'$ be a structural copy of $\mathcal{G}$ and let $v'$ be the node in $\mathcal{G}'$ which corresponds to $v$ of $\mathcal{G}$
for root node $v$ of $\mathcal{G}$, set $\mathcal{D}_v = \mathcal{D}$
**for** $v$ **in** *topdownsort(V)* **do**
    **if** *$v$ is internal* **then**
        get partition $\mathcal{X}_{i,u}$ of decision variable $X_i$ associated with $v$
        **for** $u \in \mathrm{ch}(v)$ **do**
            let $w_{v'u'} = \frac{\sum_{\mathbf{x} \in \mathcal{D}_v} \mathbb{1}(\mathbf{x}[i] \in \mathcal{X}_{i,u})}{|\mathcal{D}_v|}$
            set $\mathcal{D}_u = \{ \mathbf{x} \in \mathcal{D}_v \mid \mathbf{x}[i] \in \mathcal{X}_{i,u} \}$
        **end**
        let $v'$ be a sum node $\sum_{u' \in \mathrm{ch}(v')} w_{v'u'} u'$
    **else**
        let $v'$ be a density $p_{v'}(\mathbf{x}, y)$ with support $\mathcal{A}_v$, learned from $\mathcal{D}_v$
    **end**
**end**

**Algorithm 1:** Converting DT to PC (GeDT).

Distribution nodes are the leaves of the graph $\mathcal{G}$, while sum and product nodes are the internal nodes. Each distribution node (leaf) $v$ computes a probability density[1] over some subset $\mathbf{X}' \subseteq \mathbf{X}$, i.e. a normalised function $p_v(\mathbf{x}') \colon \boldsymbol{\mathcal{X}}' \mapsto \mathbb{R}^+$ from the state space of $\mathbf{X}'$ to the non-negative real numbers. The set of variables $\mathbf{X}'$ over which the leaf computes a distribution is called the *scope* of $v$, and denoted by $\mathrm{sc}(v) := \mathbf{X}'$. Given the scopes of the leaves, the scope of any internal node $v$ (sum or product) is recursively defined as $\mathrm{sc}(v) = \cup_{u \in \mathrm{ch}(v)} \mathrm{sc}(u)$. Sum nodes compute convex combinations over their children, i.e. if $v$ is a sum node, then $v$ computes $v(\mathbf{x}) = \sum_{u \in \mathrm{ch}(v)} w_{v,u} u(\mathbf{x})$, where $w_{v,u} \geq 0$ and $\sum_{u \in \mathrm{ch}(v)} w_{v,u} = 1$. Product nodes compute the product over their children, i.e. if $v$ is a product node, then $v(\mathbf{x}) = \prod_{u \in \mathrm{ch}(v)} u(\mathbf{x})$. The density $p(\mathbf{X})$ represented by a PC is the function computed by its root node, and can be evaluated with a feedforward pass.

The main feature of PCs is that they facilitate a wide range of *tractable* inference routines, which go hand in hand with certain structural properties, defined as follows [8, 50]: i) A sum node $v$ is called *smooth* if its children have all the same scope: $\mathrm{sc}(u) = \mathrm{sc}(u')$, for any $u, u' \in \mathrm{ch}(v)$. ii) A product node $v$ is called *decomposable* if its children have non-overlapping scopes: $\mathrm{sc}(u) \cap \mathrm{sc}(u') = \emptyset$, for any $u, u' \in \mathrm{ch}(\pi)$, $u \neq u'$. A PC is smooth (respectively decomposable) if all its sums (respectively products) are smooth (respectively decomposable). Smoothness and decomposability are sufficient to ensure *tractable marginalisation* in PCs. In particular, assume that we wish to evaluate the density over $\mathbf{X}_o \subset \mathbf{X}$ for evidence $\mathbf{X}_o = \mathbf{x}_o$, while marginalising $\mathbf{X}_{\neg o} = \mathbf{X} \setminus \mathbf{X}_o$. In PCs, this task reduces to performing marginalisation at the leaves [37], that is, for each leaf $v$ one marginalises $\mathrm{sc}(v) \cap \mathbf{X}_{\neg o}$, and evaluates it for the values corresponding to $\mathrm{sc}(v) \cap \mathbf{X}_o$. The desired marginal $p_{\mathbf{X}_o}(\mathbf{x}_o)$ results from evaluating internal nodes as in computing the complete density. Furthermore, a PC is called *deterministic* [8, 50] if it holds that for each complete sample $\mathbf{x}$, each sum node has at most one non-zero child. Determinism and decomposability are sufficient conditions for *efficient maximisation*, which again, like density evaluation and marginalisation, reduces to a single feedforward pass.

## 3   Generative Decision Trees

Given a learned DT and the dataset $\mathcal{D} = \{(\mathbf{x}_1, y_1), \ldots, (\mathbf{x}_n, y_n)\}$ it has been learned on, we can obtain a corresponding generative model, by converting the DT into a PC. This conversion is given in Algorithm 1. In a nutshell, Algorithm 1 converts each decision node into a sum node and each leaf into a density with support restricted to the leaf's cell. The training samples can be figured to be routed from the root node to the leaves, following the decisions at each decision/sum node. The sum weights are given by the fraction of samples which are routed from the sum node to each of its children. The leaf densities are learned on the data which arrives at the respective leaves.

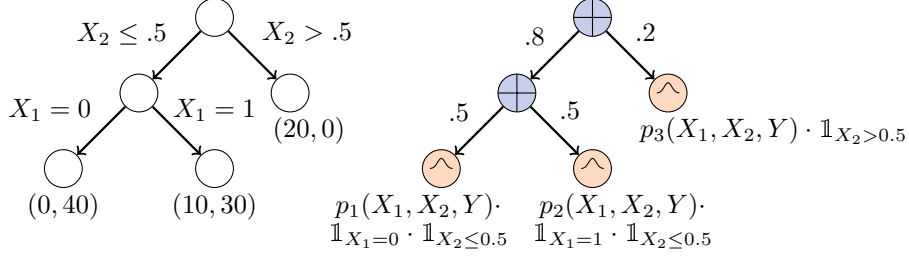

Figure 1: Illustration of a DT and its corresponding PC as obtained by Algorithm 1.

As an example, assuming $\mathbf{X}$ and $Y$ factorise at the leaves, Algorithm 1 applied to the DT on the left-hand side of Figure 1 gives the PC on the right-hand side and the following densities at the leaves:

$$p_1(X_1, X_2, Y) = p_1(X_1, X_2)(0 \cdot \mathbb{1}(Y = 0) + 1 \cdot \mathbb{1}(Y = 1)),$$
$$p_2(X_1, X_2, Y) = p_2(X_1, X_2)(0.25 \cdot \mathbb{1}(Y = 0) + 0.75 \cdot \mathbb{1}(Y = 1)),$$
$$p_3(X_1, X_2, Y) = p_3(X_1, X_2)(1 \cdot \mathbb{1}(Y = 0) + 0 \cdot \mathbb{1}(Y = 1)),$$

Note that $X_1$ is deterministic (all mass absorbed in one state) in $p_1$ and $p_2$, since $X_1$ has been fixed by the tree construction, while $p_3$ is a "proper" distribution over $X_1$ and $X_2$. Densities $p_i(X_1, X_2)$ do not appear in the DT representation and illustrate the extension brought in by the PC formalism.

We denote the output of Algorithm 1 as a *Generative Decision Tree* (GeDT). Note that GeDTs are proper PCs over $(\mathbf{X}, Y)$, albeit rather simple ones: they are tree-shaped and contain only sum nodes. They are clearly smooth, since each leaf density has the full scope $(\mathbf{X}, Y)$, and they are trivially decomposable, as they do not contain products. Thus, both the full density or any sub-marginal can be evaluated by simply evaluating the GeDT bottom up, where for marginalisation tasks we first need to perform marginalisation at the leaves. Furthermore, it is easy to show that any GeDT is *deterministic* (see supp. material). As shown in [35, 44], the sum-weights set by Algorithm 1 are in fact the *maximum likelihood* weights for deterministic PCs.

In Algorithm 1, we learn a density $p_v(\mathbf{x}, y)$ for each leaf $v$, where we have not yet specified the model or learning algorithm. Thus, we denote GeDT($M$) as a GeDT whose leaf densities are learned by "method $M$", where $M$ might be graphical models, again PCs, or even neural-based density estimators [23, 46]. In order to ensure tractable marginalisation of the overall GeDT, however, we use either *fully factorised* leaves—for each leaf $v$, $p_v(\mathbf{X}, Y) = p_v(X_1)p_v(X_2)\ldots p_v(X_m)p_v(Y)$—or PCs learned with *LearnSPN* [16]. In both cases marginalisation at the leaves, and hence in the whole GeDT, is efficient. Regardless of the model $M$, we generally learn the leaves using the maximum likelihood principle, or some proxy of it. Thus, since the sum-weights are already set to the (global) maximum likelihood solution by Algorithm 1, the overall GeDT also fits the training data. A basic design choice is how to model the dependency between $\mathbf{X}$ and $Y$ at the leaves: we might assume independence between them, i.e. assume $p(\mathbf{X}, Y) = p(\mathbf{X})p(Y)$ (*class-factorised* leaves)[2] or simply pass the data over both $\mathbf{X}$ and $Y$ to a learning algorithm and let it determine the dependency structure itself (*full* leaves). Note that we are free to have different types of density estimators for different leaves in a single GeDT. A natural design choice is to match the complexity of the estimator in a leaf to the number of samples it contains.

The main semantic difference between DTs and GeDTs is that a DT represents a classifier, i.e. a conditional distribution $f(\mathbf{x})$, while the corresponding GeDT represents a full joint distribution $p(\mathbf{X}, Y)$. The latter naturally lends itself towards classification by deriving the conditional distribution $p(Y \mid \mathbf{x}) \propto p(\mathbf{x}, Y)$. How are the original DT classifier $f(\mathbf{x})$ and the GeDT classifier $p(Y \mid \mathbf{x})$ related? In theory, $p(Y \mid \mathbf{x})$ might differ substantially from $f(\mathbf{x})$, since every feature might influence classification in a GeDT, even if it never appears in any decision node of the DT. In the case of class-factorised leaves, however, we obtain "backwards compatibility".

**Theorem 1.** *Let $f$ be a DT classifier and $p(Y \mid \mathbf{x})$ be a corresponding GeDT classifier, where each leaf in GeDT is class-factorised, i.e. of the form $p(Y)p(\mathbf{X})$, and where $p(Y)$ has been estimated in the maximum-likelihood sense. Then $f(\mathbf{x}) = p(Y \mid \mathbf{x})$, provided that $p(\mathbf{x}) > 0$.*

For space reasons, proofs and complexity results are deferred to the appendix. Theorem 1 shows that DTs and GeDTs yield exactly the same classifier for class-factorised leaves and complete data. DTs achieve their most impressive performance when used as an ensemble in RFs. It is straight-forward to convert each DT in an RF using Algorithm 1, yielding an ensemble of GeDTs. We call such an ensemble a *Generative Forest* (GeF). This result extends to ensembles, as clearly when all GeDTs in a GeF use class-factorised leaves, then according to Theorem 1, GeFs yield exactly the same prediction function as their corresponding RFs. This means that the everyday practitioner can safely replace RFs with class-factorised GeFs, gaining the ability to classify under *missing input data*.

## 4 Handling Missing Values

The probably most frequent strategy to treat missing inputs in DTs and RFs is to use some *single imputation* technique, i.e. to first predict any missing values based on the observed ones, and then use the imputed sample as input to the classifier. A particularly prominent method is *K-nearest neighbour* (KNN) imputation, which typically works well in practice. This strategy, however, is not Bayes consistent and can in principle be arbitrarily bad. This can be shown with a simple example. Assume two multivariate Gaussian features $X_1$ and $X_2$ with $var(X_1) \geq \tau$, $var(X_2) \geq \tau$ for some $\tau > 0$, i.e. the variances of $X_1$ and $X_2$ are bounded from below. Let the conditional class distribution be $p(y \mid x_1, x_2) = \mathbb{1}(|x_2 - \mathbb{E}[X_2 \mid x_1]| > \epsilon)$, i.e. $Y$ detects whether $X_2$ deviates more than $\epsilon$ from its mean, conditional on $X_1$. Assume $X_2$ is missing and use KNN to impute it, based on $X_1 = x_1$. KNN is known to be a consistent regressor, provided the number of neighbours goes to infinity but vanishes in comparison to the number of samples [9]. Thus, the imputation for $X_2$ based on $x_1$ converges to $\mathbb{E}[X_2 \mid x_1]$, yielding a constant prediction of $Y = 0$. It follows that by making $\epsilon$ arbitrarily small, we can push the classification error arbitrarily close to 1, while the true error goes to 0.

Assuming that inputs are missing at random [28] and that we have only inputs $\mathbf{x}_o$ for some subset $\mathbf{X}_o \subset \mathbf{X}$, a GeDT naturally yields a classifier $p(y \mid \mathbf{X}_o)$, by marginalising missing features as in Eq. (1). Recall that marginalisation in PCs, and thus in GeDTs, can be performed with a single feedforward pass, given that the GeDT's leaves permit efficient marginalisation. In our experiments, we use either fully factorised leaves or PC leaves learned by LearnSPN [16], a prominent PC learner, such that we can efficiently and exactly evaluate $p(Y \mid \mathbf{X}_o)$ with a single pass through the network. Thus, a GeDT represents in fact $2^{|\mathbf{X}|}$ classifiers, one for each missingness pattern. Since the true data distribution yields Bayes optimal classifiers for each $\mathbf{X}_o$, and since the parameters of GeDTs are learned in the maximum likelihood sense, using the GeDT predictor $p(y \mid \mathbf{X}_o)$ for missing data is natural. For a simplified variant of GeDTs, we can show that they converge to the true distribution and are therefore Bayes consistent classifiers for each $\mathbf{X}_o$. Theorem 2 assumes, without loss of generality, that all variables in $\mathbf{X}$ are continuous.

**Theorem 2.** *Let $\mathbb{P}^*$ be an unknown data generating distribution with density $p^*(\mathbf{X}, Y)$, and let $\mathcal{D}_n$ be a dataset drawn i.i.d. from $\mathbb{P}^*$. Let $\mathcal{G}$ be a DT learned with a DT learning algorithm, using axis-aligned splits. Let $\mathcal{A}^n$ be the (rectangular) leaf cells produced by the learning algorithm. Assume it holds that i) $\lim_{n \to \infty} |\mathcal{A}^n| \log(n)/n \to 0$ and ii) $\mathbb{P}^*(\{\mathbf{x} \mid \text{diam}(\mathcal{A}_x^n) > \gamma\}) \to 0$ almost surely for all $\gamma > 0$, where $\text{diam}(\mathcal{A})$ is the diameter of cell $\mathcal{A}$. Let $\mathcal{G}'$ be the GeDT corresponding to $\mathcal{G}$, obtained via Algorithm 1, where for each leaf $v$, $p_v$ is of the form $p_v(Y)p_v(\mathbf{X})$, with $p_v(\mathbf{X})$ uniform on $\mathcal{A}_v$ and $p_v(Y)$ the maximum likelihood Categorical (fractions of class values of samples in $\mathcal{A}_v$). Then the GeDT distribution is $l_1$-consistent, i.e. $\sum_y \int |p(\mathbf{x}, y) - p^*(\mathbf{x}, y)| \mathrm{d}\mathbf{x} \to 0$, almost surely.*

Note that the assumptions in Theorem 2 are in line with consistency results for DTs. See for example [4, 9, 30], all of which require, in some sense, that the number of cells vanishes in comparison to the number of samples, and that the cell sizes shrink to zero. Theorem 2 naturally leads to the Bayes-consistency of GeDTs and GeFs under missing inputs.

**Corollary 1.** *Under assumptions of Theorem 2, any GeDT predictor $p(Y \mid \mathbf{X}_o)$, for $\mathbf{X}_o \subseteq \mathbf{X}$ is Bayes consistent.*

**Corollary 2.** *Assume a GeF whose GeDTs are learned under assumptions of Theorem 2. Then the GeF of GeDT predictors $p(Y \mid \mathbf{X}_o)$, for any $\mathbf{X}_o \subseteq \mathbf{X}$, is Bayes consistent.*

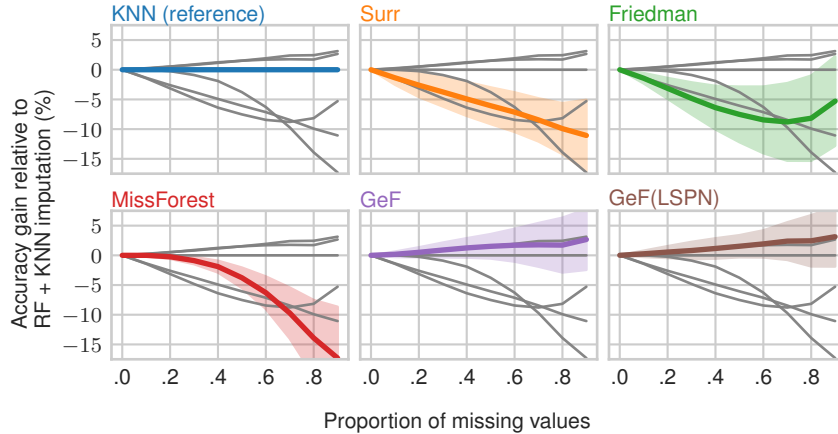

Figure 2: Average accuracy gain relative to RFs (100 trees) plus KNN imputation against proportion of missing values. The same plot is repeated six times, each time highlighting one method. The average as well as the confidence intervals (95%) are computed across the 21 datasets of Table 1.

## 5 Related Work

Among the many variations of DTs and RFs that have been proposed in the last decades, the closest to our work are those that, similarly to GeDTs and GeFs, extend DT leaves with "non-trivial" models. Notable examples are DTs where the leaves are modelled by linear and logistic regressors [13, 26, 43], kernel density estimators (KDEs) [29, 47], linear discriminant models [15, 22], KNN classifiers [5, 29], and Naive-Bayes classifiers (NBCs) [25]. Nonetheless, all these previous works focus primarily on improving the classification accuracy or smoothing probability estimates but do not model the full joint distribution, like in this work. Even extensions by Smyth et al. [47] and Kohavi [25], which include generative models (KDEs and NBCs, respectively) do not exploit their generative properties. To the best of our knowledge, GeFs are the first DT framework that effectively model and leverage the full joint distribution in a classification context. That is of practical significance as none of these earlier extensions of DTs offer a principled way to treat missing values or detect outliers. Here it is also worth mentioning the contemporary work of Khosravi et al. [21] that proposes a similar probabilistic approach to handle missing data in DTs.

On the other side of the spectrum, DTs have also been extended to density estimators [18, 44, 45, 53]. Among these, Density Estimation Trees (DETs) [45], Cutset Networks (CNets) [44], and randomised ensembles thereof [10], are probably the closest to our work. These models are trained with a greedy tree-learning algorithm but minimise a modified loss function that matches their generative nature: joint entropy across all variables in CNets, mean integrated squared error in DETs. Notably, CNets, like GeFs, are probabilistic circuits, and hence also allow for tractable inference and marginalisation. They, however, have not been applied in a discriminative setting and are not backwards compatible with DTs and RFs. Moreover, GeDTs (and GeFs) can be seen as a family of models depending on the estimation at the leaves, making a clear parallel with what DTs (and RFs) offer.

Finally, one can also mimic the benefits of generative models in ensembles by learning predictors for all variables, as in MERCS [52]. That is fundamentally different from our probabilistic approach and might entail prohibitively large numbers of predictors. Handling missing values, in the worst case, would require one predictor for each of the $2^{|\mathbf{X}|}$ missing patterns, and that is why MERCS relies on imputation methods when needed. Conversely, GeDTs model a full joint distribution, thus being more compact and interpretable.

## 6 Experiments

We run a series of classification tasks with incomplete data to compare our models against surrogate splits [4, 49], Friedman's method [14, 41], and mean (mode), KNN ($k = 7$) and MissForest [48] imputation. In particular, we experiment with two variants of GeFs: one with fully-factorised leaves,

Table 1: Accuracy at 30% percent of missing values at test time with 95% confidence intervals. The best performing model is underlined, whereas all models within its confidence interval appear in bold.

| Dataset | n | Surrogate | Friedman | Mean | KNN | MissForest | GeF | GeF(LSPN) |
|---|---|---|---|---|---|---|---|---|
| dresses | 500 | 45.48 ± 1.57 | 55.8 ± 1.21 | $\underline{\mathbf{58.18}}$ ± .85 | 56.62 ± 1.57 | 55.68 ± .95 | 57.12 ± 1.11 | 57.14 ± 1.07 |
| wdbc | 569 | 94.96 ± .36 | 94.96 ± .35 | 94.92 ± .58 | 95.59 ± .41 | 94.92 ± .36 | 95.64 ± .47 | $\underline{\mathbf{96.26}}$ ± .42 |
| diabetes | 768 | 72.97 ± .73 | **73.35** ± .70 | 71.67 ± .84 | 72.4 ± .75 | 72.46 ± .92 | $\underline{\mathbf{73.93}}$ ± .63 | **73.83** ± .72 |
| vehicle | 846 | **71.61** ± .92 | 67.12 ± .79 | 63.27 ± 1.05 | **71.77** ± 1.01 | 70.69 ± 1.33 | **72.39** ± 1.13 | $\underline{\mathbf{72.77}}$ ± 1.27 |
| vowel | 990 | 78.79 ± .86 | 70.81 ± 1.45 | 64.51 ± .83 | 85.62 ± .66 | 81.85 ± 1.10 | **89.25** ± .77 | $\underline{\mathbf{89.59}}$ ± .91 |
| credit-g | 1000 | 71.97 ± .32 | 72.42 ± .31 | 73.01 ± .64 | 73.06 ± .65 | 73.03 ± .78 | **73.81** ± .38 | $\underline{\mathbf{73.97}}$ ± .36 |
| mice | 1080 | 95.84 ± .44 | 91.01 ± .77 | 84.91 ± 1.03 | 97.7 ± .50 | 96.08 ± .52 | 98.38 ± .32 | $\underline{\mathbf{99.06}}$ ± .15 |
| authent. | 1372 | 88.65 ± .74 | 87.13 ± .69 | 84.47 ± .79 | $\underline{\mathbf{91.98}}$ ± .55 | 90.74 ± .37 | 90.33 ± .65 | 89.66 ± .64 |
| cmc | 1473 | 48.7 ± .79 | **49.8** ± .38 | 47.67 ± .86 | 48.38 ± .58 | 48.28 ± .90 | **49.96** ± 1.03 | $\underline{\mathbf{50.08}}$ ± 1.05 |
| segment | 2310 | 93.32 ± .27 | 84.14 ± .69 | 78.34 ± .68 | $\underline{\mathbf{94.25}}$ ± .32 | 93.21 ± .41 | 93.42 ± .20 | 93.41 ± .34 |
| dna | 3186 | **90.53** ± .31 | 77.23 ± .36 | 83.91 ± .43 | 89.31 ± .33 | $\underline{\mathbf{90.76}}$ ± .34 | 87.42 ± .19 | 82.99 ± .25 |
| splice | 3190 | 86.09 ± .46 | 84.76 ± .65 | 84.65 ± .28 | 89.06 ± .53 | 86.18 ± .26 | $\underline{\mathbf{91.1}}$ ± .50 | 85.69 ± .53 |
| krvskp | 3196 | 73.62 ± .92 | 82.81 ± .73 | 83.58 ± .64 | 86.58 ± .43 | 86.24 ± .56 | $\underline{\mathbf{88.35}}$ ± .39 | **88.65** ± .32 |
| robot | 5456 | 91.74 ± .23 | 84.73 ± .57 | 89.39 ± .28 | 92.74 ± .25 | 91.72 ± .37 | 92.97 ± .28 | $\underline{\mathbf{94.67}}$ ± .20 |
| texture | 5500 | 95.31 ± .15 | 89.85 ± .34 | 84.24 ± .42 | $\underline{\mathbf{97.13}}$ ± .15 | 95.4 ± .17 | 95.93 ± .15 | **97.12** ± .13 |
| wine | 6497 | 84.49 ± .17 | 82.45 ± .10 | 83.2 ± .15 | 85.73 ± .26 | $\underline{\mathbf{85.95}}$ ± .12 | 85.22 ± .18 | **85.85** ± .19 |
| gesture | 9873 | 58.37 ± .15 | 52.86 ± .27 | 55.41 ± .21 | $\underline{\mathbf{61.62}}$ ± .22 | **61.48** ± .26 | 58.65 ± .18 | 60.2 ± .21 |
| phishing | 11055 | 81.52 ± .50 | 88.98 ± .17 | 88.02 ± .19 | 92.06 ± .13 | 91.18 ± .18 | 92.99 ± .08 | $\underline{\mathbf{93.3}}$ ± .06 |
| bank | 41188 | 90.42 ± .18 | 90.3 ± .15 | 90.09 ± .11 | **90.64** ± .14 | 90.4 ± .19 | $\underline{\mathbf{90.79}}$ ± .21 | **90.77** ± .21 |
| jungle | 44819 | 63.45 ± .24 | 71.91 ± .12 | 66.89 ± .40 | 66.25 ± .15 | 65.67 ± .20 | $\underline{\mathbf{72.4}}$ ± .12 | 72.3 ± .11 |
| electricity | 45312 | 79.79 ± .09 | 77.47 ± .10 | 73.24 ± .19 | 80.55 ± .11 | 81.21 ± .10 | 82.23 ± .12 | $\underline{\mathbf{82.64}}$ ± .11 |

which we denote simply GeF, and another with leaves learned via LearnSPN [16], which we call GeF(LearnSPN). We use a transformation of GeFs into a clever PC that prunes unnecessary sub-trees [6], speeding up computations and achieving time complexity comparable to the original DTs and RFs (see supp. material). In all experiments, GeF, GeF(LearnSPN) and the RF share the exact same structure (partition over the feature space) and are composed of 100 trees; including more trees has been shown to yield only marginal gains in most cases [39]. In GeF(LearnSPN), we run LearnSPN only for leaves with more than 30 samples, defaulting to a fully factorised model in smaller leaves.

We compare the accuracy of the methods in a selection of datasets from the OpenML-CC18 benchmark[3] [51] and the wine-quality dataset [33]. Table 1 presents results for 30% of missing values at test time (different percentages are shown in the supp. material), with 95% confidence intervals across 10 repetitions of 5-fold cross-validation. GeF models outperform other methods in almost all datasets, validating that the joint distributions at the leaves provide enough information for computing the marginalisation in Eq. (1). We also note that increasing the expressive power of the models at the leaves seems worthwhile, as GeF(LSPN) outperforms the vanilla GeF in about half of the datasets. Similar conclusions are supported by Figure 2, where we plot the average gain in accuracy relative to RF + KNN imputation at different proportions of missing values. While earlier built-in methods, Friedman's and surrogate splits, perform poorly (justifying the popularity of imputation techniques for RFs), GeFs are on average more than 3% more accurate than KNN imputation. For the sake of space, a thorough exposition of these experiments is deferred to the supp. material, where we fully describe the experimental procedure, show different percentages of missing data and include results with PCs learned via *class-selective* LearnSPN [6], as baseline for a standard generative model.

The reviewers suggested a direct comparison against CNets [44] since, like GeFs, they are based on DTs and encode a proper joint distribution over all the variables. However, while we acknowledge the value of such comparison, the implementations to which we had access either did not support missing data or were too slow to yield reliable experimental results with ensembles of similar size, in the short time available to revise the paper. Also, it is worth noticing that CNets are currently not available for mixed variables, which prevents their application to most datasets in Table 1.

Another advantage of generative models is the ability of using the likelihood over the explanatory variables to detect outliers. GeFs are still an ensemble of generative GeDTs and thus do not encode a single full joint distribution. However, we can extend GeFs to model a single joint by considering a uniform mixture of GeDTs (using a sum node), instead of an ensemble of the conditional distributions of each GeDT. In this case, the model represents the joint $p(\mathbf{X}, Y) = n_t^{-1} \sum_{j=1}^{n_t} p_j(\mathbf{X}, Y)$, where each $p_j$ comes from a different GeDT. This model is named GeF$^+$ and achieves similar but slightly inferior performance than GeFs in classification with missing data (still clearly superior to KNN

imputation). This does not come as a surprise: the benefits of a fully generative models often comes at the cost of a (small) drop in classification accuracy (results in the supp. material).

We illustrate how to detect outliers with GeFs by applying a GeF$^+$(LSPN) to the the wine dataset [7] with a variant of transfer testing [2]. We learn two different GeF$^+$(LSPN) models, each with only one type of wine data (red or white), to predict whether a wine has a score of 6 or higher. We then compute the log-density of unseen data (70/30 train test split) for the two wine types with both models. As we see in the histograms of Figure 3, the marginal distribution over explanatory variables does provide a strong signal to identify out-of-domain instances. In comparison to a Gaussian Kernel Density Estimator (KDE), GeF$^+$(LSPN) achieved similar results even though its structure has been fit in a discriminative way.

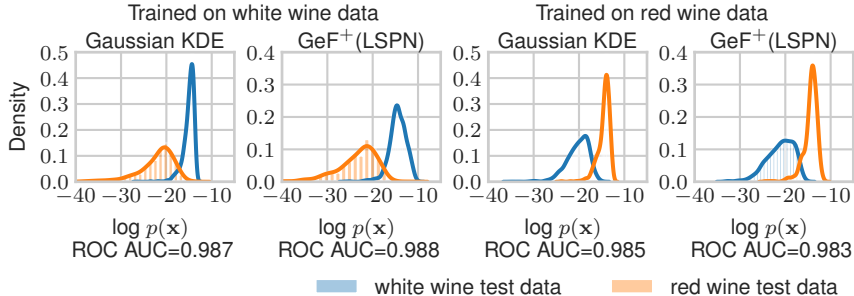

Figure 3: Normalised histograms of $\log p(\mathbf{x})$ for samples from two different wine datasets.

We repeat a similar experiment with images, where we use the MNIST dataset [27] to fit a Gaussian KDE, a Random Forest and its corresponding GeF$^+$. We then evaluate these models on different digit datasets, namely Semeion [11] and SVHN [34] (converted to grayscale and 784 pixels), to see whether they can identify out-of-distribution samples. We also use the entropy over the class variable as a baseline, since this is a signal that is easily computed on a standard Random Forest. Again, GeF$^+$ successfully identified out-of-domain samples, outperforming the two other methods and even encoding slightly different distributions for SVHN and Semeion digits. Note that in both experiments we also compare the methods in terms of the area under the receiver operating characteristic curve (AUC ROC), which we computed using the log-density (or entropy) as a signal for a binary classifier that discriminates between in- and out-of-domain samples.

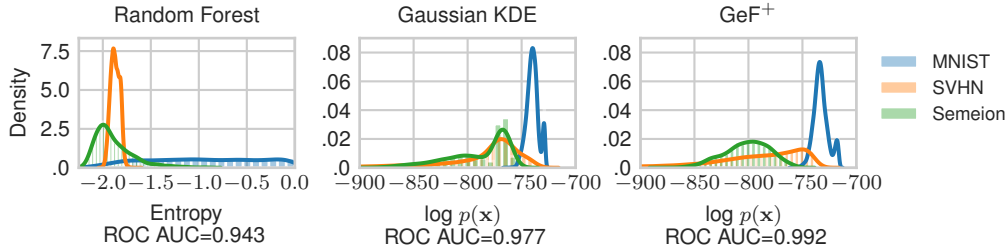

Figure 4: Normalised histograms of $\log p(\mathbf{x})$ for samples from three different image datasets.

## 7 Conclusion

By establishing a connection between Decision Trees (DTs) and Probabilistic Circuits (PCs), we have upgraded DTs to a full joint model over both inputs and outputs, yielding their generative counterparts, called GeDTs. The fact that GeDTs, and their ensemble version GeFs, are "backwards compatible" to DTs and RFs, while offering benefits like consistent classification under missing inputs and outlier detection, makes it easy to adopt them in everyday practice. Missing data and outliers, however, are just the beginning. We believe that many of the current challenges in machine learning, like explainability, interpretability, and (adversarial) robustness are but symptoms of an overemphasis on purely discriminative methods in the past decades, and that hybrid generative approaches—like the one in this paper—will contribute significantly towards mastering these current challenges.

## Broader Impact

This work establishes a connection between two sub-fields in machine learning, namely decision trees/random forests and probabilistic circuits. Since there was very restricted communication between these two research communities, a fruitful cross-fertilisation of ideas, theory and algorithms between these research domains can be expected. This represents a highly positive impact on fundamental machine learning and artificial intelligence research.

Decision trees and random forests are a de facto standard classification and regression tools in daily applied machine learning and data science. Being—so far—purely discriminative models, they struggle with two problems which are key concerns in this work: missing data and outlier detection. Since the improvements suggested in this paper can be incorporated in existing decision tree algorithms with very minor changes, our results have a potentially dramatic and immediate impact on a central and widely used machine learning and data science tool.

Since our work is elementary machine learning research, its ethical consequences are hard to assess. However, the main ethical and societal impact of our work is the extension of a standard prediction tool, increasing its application domain and pertinence, and thus amplifying existing ethical considerations of data-driven and automatic prediction.

## Acknowledgments and Disclosure of Funding

The authors thank the reviewers for their useful insights and suggestions. During part of the three years prior to the submission of this work, the authors were affiliated with the following institutions besides TU Eindhoven: Alvaro Correia was a full-time employee at Accenture and Itaú-Unibanco, and affiliated with Utrecht University; Cassio de Campos was affiliated with Queen's University Belfast and Utrecht University; Robert Peharz was affiliated with the University of Cambridge.

## Footnotes

[1]By an adequate choice of the underlying measure, this also subsumes probability mass functions.

[2]Note that such independence is only a context-specific one, conditional on the state of variables associated with sum nodes [36, 38]. This assumption does not represent global independence between $\mathbf{X}$ and $Y$.

[3]https://www.openml.org/s/99/data

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
