[Supplementary Material]

# Supplementary Material for the Paper
# "Joints in Random Forests"

**Alvaro H. C. Correia**
a.h.chaim.correia@tue.nl
Eindhoven University of Technology

**Robert Peharz**
r.peharz@tue.nl
Eindhoven University of Technology

**Cassio de Campos**
c.decampos@tue.nl
Eindhoven University of Technology

## A  Theoretical Results

**Proposition 1.** *A GeDT is* deterministic.

*Proof.* Consider any sum node $v$ in a GeDT and assume, for simplicity, that it has two children $u'$ and $u''$. Node $v$ is associated with a partition $\{\boldsymbol{\mathcal{X}}_{u'}, \boldsymbol{\mathcal{X}}_{u''}\}$ of $\boldsymbol{\mathcal{X}}_v$. Any leaf $l$ which is a descendant of $u'$, respectively $u''$, must have a support which is a subset of $\boldsymbol{\mathcal{X}}_{u'}$, respectively $\boldsymbol{\mathcal{X}}_{u''}$. Assume that $u'(\mathbf{x}) > 0$ for certain $\mathbf{x}$, implying $\mathbf{x} \in \boldsymbol{\mathcal{X}}_{u'}$ and thus $\mathbf{x} \notin \boldsymbol{\mathcal{X}}_{u''}$. Therefore, $u''(\mathbf{x}) = 0$, since $\mathbf{x}$ is not in the support of any leaf below $u''$. The same argument holds for the reverse case and straightforwardly extends to arbitrarily many sum nodes. Thus $v$ is deterministic. $\square$

**Theorem 1.** *Let $f$ be a DT classifier and $p(Y \,|\, \mathbf{x})$ be a corresponding GeDT classifier, where each leaf in GeDT is class-factorised, i.e. of the form $p(Y)p(\mathbf{X})$, and where $p(Y)$ has been estimated in the maximum-likelihood sense. Then $f(\mathbf{x}) = p(Y \,|\, \mathbf{x})$, provided that $p(\mathbf{x}) > 0$.*

*Proof.* Recall that the leaves in the GeDT are in one-to-one correspondence with the leaf cells $\mathcal{A}$ of the DT, and that the support of any leaf is given by its corresponding $\mathcal{A} \in \mathcal{A}$. Let $v_{\mathbf{x}}$ be the unique leaf in the GeDT whose cell is $\mathcal{A}(\mathbf{x})$. Since GeDT is a tree-shaped PC containing only sum nodes, its joint distributions is either $p_{v_{\mathbf{x}}}(\mathbf{x}, y)$—if GeDT consists only of $v_{\mathbf{x}}$—or can be written as

$$p(\mathbf{x}, y) = \sum_{u \in \mathrm{ch}(v)} w_{v,u} u(\mathbf{x}), \tag{1}$$

where $v$ is the root node. Since the GeDT is deterministic, it has at most one non-zero child. From $p(\mathbf{x}) > 0$ it follows that the GeDT has *exactly* one non-zero child, say $u'$, and (1) can be written as $p(\mathbf{x}, y) = w_{v,u'} u'(\mathbf{x}, y)$. Now, since $u'(\mathbf{x}, y)$ is also a tree-shape PC containing only sums, it follows by induction that $p(\mathbf{x}, y) = \left(\prod_{(v,u) \in \Lambda} w_{v,u}\right) p_{v_{\mathbf{x}}}(\mathbf{x}, y)$, where $\Lambda$ is the unique path from root to $v_{\mathbf{x}}$ following only non-zero nodes, and $w_{v,u}$ are the sum-weights of edges $(v, u)$ in $\Lambda$. Since each leaf is class-factorised, we have $p_{v_{\mathbf{x}}}(\mathbf{x}, y) = p_{v_{\mathbf{x}}}(\mathbf{x}) p_{v_{\mathbf{x}}}(y)$, and $\left(\prod_{(v,u) \in \Lambda} w_{v,u}\right) p_{v_{\mathbf{x}}}(\mathbf{x}) p_{v_{\mathbf{x}}}(y) \propto p(y \,|\, \mathbf{x}) = p_{v_{\mathbf{x}}}(y) = f^{\mathcal{A}(\mathbf{x})}(\mathbf{x}) = f(\mathbf{x})$, since each $f^{\mathcal{A}}(\mathbf{x})$ is—like $p_{v_{\mathbf{x}}}$—learned by the class proportions of samples falling in $\mathcal{A}$. $\square$

**Theorem 2.** *Let $\mathbb{P}^*$ be an unknown data generating distribution with density $p^*(\mathbf{X}, Y)$, and let $\mathcal{D}_n$ be a dataset drawn i.i.d. from $\mathbb{P}^*$. Let $\mathcal{G}$ be a DT learned with a DT learning algorithm, using axis-aligned splits. Let $\mathcal{A}^n$ be the (rectangular) leaf cells produced by the learning algorithm. Assume it holds that i) $\lim_{n \to \infty} |\mathcal{A}^n| \log(n)/n \to 0$ and ii) $\mathbb{P}^*(\{\mathbf{x} \,|\, \mathrm{diam}(\mathcal{A}_{\mathbf{x}}^n) > \gamma\}) \to 0$ almost surely for all $\gamma > 0$, where $\mathrm{diam}(\mathcal{A})$ is the diameter of cell $\mathcal{A}$. Let $\mathcal{G}'$ be the GeDT corresponding to $\mathcal{G}$, obtained*

*via Algorithm 1, where for each leaf $v$, $p_v$ is of the form $p_v(Y)p_v(\mathbf{X})$, with $p_v(\mathbf{X})$ uniform on $\mathcal{A}_v$ and $p_v(Y)$ the maximum likelihood Categorical (fractions of class values of samples in $\mathcal{A}_v$). Then the GeDT distribution is $l_1$-consistent, i.e. $\sum_y \int |p(\mathbf{x}, y) - p^*(\mathbf{x}, y)| d\mathbf{x} \to 0$, almost surely.*

Before proving Theorem 2 we need to introduce some background. This theorem extends consistency results for collections of partitions of the state space $\mathcal{X}$, as discussed by Lugosi and Nobel [12]. A central notion is the *growth function* of such partitions.

**Definition 1** (Growth function [12])**.** *Let $\mathcal{X}$ be some set and $\mathcal{F}$ be a collection of finite partitions of $\mathcal{X}$. Let $\boldsymbol{\xi} = \{\xi_1, \ldots, \xi_n\}$ be a set of points from $\mathcal{X}$. Let $\Delta(\mathcal{F}, \boldsymbol{\xi})$ be the number of distinct partitions induced by $\mathcal{F}$, that is the size of set $\{\{\boldsymbol{\xi} \cap \mathcal{A} \mid \mathcal{A} \in \mathcal{A}\} \mid \mathcal{A} \in \mathcal{F}\}$. The growth function is defined as $\Delta^*(\mathcal{F}) = \sup_{\boldsymbol{\xi}} \Delta(\mathcal{F}, \boldsymbol{\xi})$, where the $\sup$ ranges over all sets of $n$ points from $\mathcal{X}$.*

Note that the growth function $\Delta^*$ is defined akin to the *dichotomic growth function*, as introduced by Vapnik and Chervonenkis and well known in statistical learning theory [22]. In particular, we derive the following bound of $\Delta^*$.

**Proposition 2.** *Let $\mathcal{X}$ be some set and $\mathcal{C}$ be any collection of subsets of $\mathcal{X}$. Let $\Phi(\mathcal{C}, \boldsymbol{\xi})$ be the shatter coefficient of point set $\boldsymbol{\xi}$ and $\Phi^*(\mathcal{C}) = \sup_{\boldsymbol{\xi}} \Phi(\mathcal{C}, \boldsymbol{\xi})$ be the dichotomic growth function [22]. Let $\mathcal{F}$ be a collection of finite partitions of $\mathcal{X}$, as in Definition 1, where the maximal partition size is $J := \sup_{\mathcal{A} \in \mathcal{F}} |\mathcal{A}|$. If $\mathcal{C} = \{\mathcal{A} \mid \mathcal{A} \in \mathcal{A}, \mathcal{A} \in \mathcal{F}\}$ then*

$$\Delta(\mathcal{F}, \boldsymbol{\xi}) \leq \Phi(\mathcal{C}, \boldsymbol{\xi})^J, \tag{2}$$

*and moreover $\Delta^*(\mathcal{F}) \leq \Phi^*(\mathcal{C})^J$.*

*Proof.* Let the point set $\boldsymbol{\xi}$ be fixed. Any partition $\{\boldsymbol{\xi} \cap \mathcal{A} \mid \mathcal{A} \in \mathcal{A}\}$, for some $\mathcal{A} \in \mathcal{F}$, can be written as $\{\boldsymbol{\xi} \cap \mathcal{A}_1, \ldots, \boldsymbol{\xi} \cap \mathcal{A}_J\}$ for some $\mathcal{A}_1, \ldots, \mathcal{A}_J \in \mathcal{C}$, since $\mathcal{C}$ contains all cells which appear in $\mathcal{F}$. Thus, $\Delta(\mathcal{F}, \boldsymbol{\xi}) \leq |\{\{\boldsymbol{\xi} \cap \mathcal{A}_1, \ldots, \boldsymbol{\xi} \cap \mathcal{A}_J\} \mid \mathcal{A}_1, \ldots, \mathcal{A}_J \in \mathcal{C}\}|$. Note that the number of partitions of this form is bounded by

$$|\{\{\boldsymbol{\xi} \cap \mathcal{A}_1, \ldots, \boldsymbol{\xi} \cap \mathcal{A}_J\} \mid \mathcal{A}_1, \ldots, \mathcal{A}_J \in \mathcal{C}\}| \leq \bigtimes_{j=1}^J |\{\boldsymbol{\xi} \cap \mathcal{A}_j \mid \mathcal{A}_j \in \mathcal{C}\}|. \tag{3}$$

The right hand side of (3) is $\Phi(\mathcal{C}, \boldsymbol{\xi})^J$, and thus (2) follows. $\Delta^*(\mathcal{F}) \leq \Phi^*(\mathcal{C})^J$ follows from applying $\sup_{\boldsymbol{\xi}}$ on both sides of (2). $\square$

In our case, we study partitions $\mathcal{A}$ induced by a DT, each of which divides $\mathcal{X}$ into a set of hyper-rectangles.[1] Hence, we consider the collection of partitions $\mathcal{F}$ containing all possible partitions whose sets are hyper-rectangles. We are now ready to prove Theorem 2.

*Proof.* Let $\mathcal{F}^n$ be the collection of all DT partitions which can be generated for sample size $n$, i.e. $\mathcal{A}^n \in \mathcal{F}^n$. By Proposition 2, we know that $\Delta^*(\mathcal{F}^n) \leq \Phi^*(\mathcal{C})^{|\mathcal{A}^n|}$, where $\mathcal{C}$ is the collection of all sub-rectangles in $\mathcal{X}$. The VC dimension [22] of $\mathcal{C}$ is known to be $2|\mathbf{X}|$, and consequently, by Sauer's lemma, $\Delta^*(\mathcal{F}) \leq \Phi^*(\mathcal{C})^{|\mathcal{A}^n|} \leq Cn^{2|\mathcal{A}^n||\mathbf{X}|}$, where $C$ is a constant depending only on $|\mathbf{X}|$. Therefore, if condition i) holds ($\lim_{n \to \infty} |\mathcal{A}^n| \log(n)/n \to 0$) it follows that $\frac{\log \Delta^*}{n} \to 0$. Thus, together with condition ii) all conditions of Theorems 1 and 2 in [12] hold.

Since the GeDT is deterministic, its distribution can be written as

$$p(\mathbf{x}, y) = \left( \prod_{(v, u) \in \Lambda} w_{v, u} \right) p_{v_{\mathbf{x}}}(\mathbf{x}, y), \tag{4}$$

where $v_{\mathbf{x}}$ is the unique non-zero leaf in the GeDT, $\Lambda$ is the unique path from the root to $v_{\mathbf{x}}$ following only non-zero nodes, and $w_{v, u}$ are the sum-weights of edges $(v, u)$ in $\Lambda$ (see also proof of Theorem 1).

It is easy to see that $\prod_{(v, u) \in \Lambda} w_{v, u} = \hat{\mathbb{P}}(\mathcal{A}_{\mathbf{x}})$, where $\hat{\mathbb{P}}$ is the empirical distribution of $\mathcal{D}_n$, i.e. the fraction of data points falling in $\mathcal{A}_{\mathbf{x}}$ (see Algorithm 1 in the main paper). The distribution computed

by each leaf $v$ is, by assumption, $p_v(\mathbf{x}, y) = p_v(y)\frac{1}{\text{vol}(\mathcal{A}_\mathbf{x})}$, where $\text{vol}(\mathcal{A})$ is the volume (Lebesgue measure) of $\mathcal{A}$. Thus, we can write (4) as

$$p(\mathbf{x}, y) = p_v(y)\hat{\mathbb{P}}(\mathcal{A}_\mathbf{x})\frac{1}{\text{vol}(\mathcal{A}_\mathbf{x})}. \tag{5}$$

By Theorem 1 in [12], $\hat{\mathbb{P}}(\mathcal{A}_\mathbf{x})\frac{1}{\text{vol}(\mathcal{A}_\mathbf{x})}$ converges to $p^*(\mathbf{x})$, while by Theorem 2 in [12], $p_v(y)$ converges to $p^*(y \mid \mathbf{x})$, both in $l1$-sense. Clearly both factors, $\hat{\mathbb{P}}(\mathcal{A}_\mathbf{x})\frac{1}{\text{vol}(\mathcal{A}_\mathbf{x})}$ and $p_v(y)$, have bounded $l1$-norm. Thus, their product converges to $p^*(y \mid \mathbf{x})p^*(\mathbf{x}) = p^*(y, \mathbf{x})$, which concludes the proof. $\square$

**Corollary 1.** *Under assumptions of Theorem 2, any GeDT predictor $p(Y \mid \mathbf{X}_o)$, for $\mathbf{X}_o \subseteq \mathbf{X}$ is Bayes consistent.*

*Proof.* Since $p(y, \mathbf{x})$ converges almost surely to $p^*(y, \mathbf{x})$ in $l1$-sense, it gives rise to the Bayes optimal classifier $\arg\max_y p^*(y, \mathbf{x})$. Consider any $X_i \in \mathbf{X}$. The marginal distribution, $X_i$ marginalised out, is $\int p(y, \mathbf{x}_{\neg i}, x_i)\mathrm{d}x_i$. Since

$$\int |p(y, \mathbf{x}_{\neg i}) - p^*(y, \mathbf{x}_{\neg i})|\mathrm{d}\mathbf{x}_{\neg i} = \int \left| \int p(y, \mathbf{x}_{\neg i}, x_i) - p^*(y, \mathbf{x}_{\neg i}, x_i)\mathrm{d}x_i \right| \mathrm{d}\mathbf{x}_{\neg i} \tag{6}$$

$$\leq \int |p(y, \mathbf{x}) - p^*(y, \mathbf{x})|d\mathbf{x}, \tag{7}$$

also the marginal converges in $l1$-sense to the true $p^*(y, \mathbf{x}_{\neg i})$. By repeating the argument, every sub-marginal converges, and thus gives rise to the corresponding Bayes optimal classifier.

$\square$

**Corollary 2.** *Assume a GeF whose GeDTs are learned under assumptions of Theorem 2. Then the GeF of GeDT predictors $p(Y \mid \mathbf{X}_o)$, for any $\mathbf{X}_o \subseteq \mathbf{X}$, is Bayes consistent.*

*Proof.* This follows directly from Proposition 1 in [1], whereby if a sequence of classifiers is Bayes-consistent, then the classifier obtained by averaging them is also consistent. $\square$

# B Time Complexity

Let $n$ be the total number of samples and $m$ the total number of features. Regarding the learning algorithm, a Random Forest and its corresponding PC only differ in the distributions at leaves, which use a partition of the data. Therefore, assuming a tree is grown as in [2] with $\lceil m/c \rceil$ features considered at each split ($c$ a positive natural), structure learning in both models has worst-case asymptotic complexity of $\mathcal{O}(mr\,n\log n)$, where $r \in \mathcal{O}(n)$ is the number of internal nodes in the obtained tree [11]. For GeDTs, however, there is the additional cost of learning a distribution at each leaf. If $q(m)$ is the worst-case cost of the leaf learner for a constant amount of data, then the overall time complexity (for learning all leaves) is $\mathcal{O}(r\,q(m))$.

Nonetheless, if the leaf learner is such that $q(m) \leq \mathcal{O}(mn\log n)$, then the complexity is dominated by the structure learning and Random Forests and GeFs have the same worst-case asymptotic complexity of $\mathcal{O}(n_t\,(mr\,n\log n + rq(m))) \leq \mathcal{O}(n_t\,mr\,n\log n)$, where $n_t$ is the number of trees in the model. Note that $q(m) \leq \mathcal{O}(mn\log n)$ holds for many learning algorithms when only a small number of training samples fall in each leaf—namely, LearnSPN and fully-factorised leaves—provided the reasonable assumption that $m$ is $\mathcal{O}(n)$.

Figure 1: Illustration of pulling indicators up to speed up computations (in the example, $\mathbf{X}$ and $Y$ factorise at leaves). On top, the original decision tree (DT) is shown. Below, both models represent the probabilistic circuit for the original DT and encode the very same distribution, even if the one in the right-hand side is not decomposable.

To perform inference for a complete test sample, GeDTs require traversing the whole structure once (hence time $\mathcal{O}(r)$), while DTs have a worst-case of $\mathcal{O}(d)$, where $d$ is the height of the tree. However, we can bring the complexity of GeDTs down to $\mathcal{O}(d)$ by placing the indicators that define the decisions of the internal nodes of the DT near the corresponding internal nodes of the GeDT. This requires augmenting GeDTs with product nodes, one for each internal sum node. Every new product node has two children: a sum node and an indicator mimicking the decision tree split, that is, the indicator only evaluates to one if that path in the tree is active. Figure 1 illustrates the idea using the running example of the main paper, where the densities are as follows

$$
\begin{aligned}
p_1(X_1, X_2, Y) &= p_1(X_1, X_2)(0 \cdot \mathbb{1}(Y = 0) + 1 \cdot \mathbb{1}(Y = 1)), \\
p_2(X_1, X_2, Y) &= p_2(X_1, X_2)(0.25 \cdot \mathbb{1}(Y = 0) + 0.75 \cdot \mathbb{1}(Y = 1)), \\
p_3(X_1, X_2, Y) &= p_3(X_1, X_2)(1 \cdot \mathbb{1}(Y = 0) + 0 \cdot \mathbb{1}(Y = 1)).
\end{aligned}
$$

This idea does not change results, since it is the same as bringing the common indicators that appeared in the leaves of a sub-tree up towards the root of that sub-tree using the distributive property of

multiplication (for the PC enthusiast, the lack of decomposability is tackled by the determinism of the indicators). By evaluating indicators as soon as possible in a top-down recursive computation, we can avoid computing all sub-trees for which a zero is returned to a product node. With this type of computational graph, GeFs and RFs have a similar inference procedure. Predicting the class of an instance amounts to traversing each tree and evaluating the corresponding leaf, and thus the inference complexity is $\mathcal{O}(n_t d)$.

For incomplete data, however, GeDTs need to reach every active leaf (just as Friedman's method). Assuming the number of missing values in each instance is bounded by a constant, GeFs still take time $\mathcal{O}(n_t d)$, being faster than Random Forests with KNN imputation, which in the worst case take time $\mathcal{O}(n_t d + nm)$. For large (non-constant) percentages of missing values, GeFs can be as slow as $\mathcal{O}(n_t r)$ (as it may need to reach all leaves). In this case of large numbers of missing values per instance, GeFs are faster than Random Forests with KNN imputation if $d \approx r$ but slower if $d \ll r$.

# C   Missing Values Experimental Results

All 21 datasets are listed here in alphabetical order. For each of them, we report (both in tabular and graphic formats) the accuracy values at different percentages of missing data at test time, with 95% confidence intervals. These confidence intervals are computed across 10 repetitions of 5-fold cross validation with different random seeds. The datasets were obtained directly from the OpenML-CC18 benchmark web-page [2] [21], and the only pre-processing step was standardising continuous features (mean $\mu = 0$ and standard deviation $\sigma = 1$) and mapping categorical features to $\{0, \ldots, K_i - 1\}$. The datasets as well as the source code are provided with the supp. material.

We also present a few relevant details of each dataset.

> n: number of samples.
>
> $m_0$: number of categorical variables.
>
> $m_1$: number of numerical variables.
>
> $|\mathcal{Y}|$: number of classes.
>
> %Maj: percentage of the majority class.

For the sake of completeness, we briefly discuss each of the methods and their implementations. The source code is all in Python 3 and all experiments were run in a single laptop with a modern CPU.

**Random Forest implementation**   In all experiments, the structure of all models is kept the same, that is, they are all derived from the same Random Forest and thus share the same partition of the feature space. For every dataset, the Random Forests were composed of 100 "deep" trees, that is, the only stop criterion is the impurity of the class variable, possibly leading to many leaves with a single sample. Each tree is learned on a bootstrap sample of the same size of the training dataset, and each split only evaluates $\sqrt{m}$ variables, with $m$ the total number of features. We use the Gini impurity measure as the criterion to select the best split in the decision-tree learning and rank surrogate splits according to how well they predict the best split, as in [20]. The trees are all binary, with splits on categorical variables defined by two subsets of the possible instantiations. That is somewhat different from other implementations, where the splits are either "full", yielding one child per category, or given by a threshold, which implicitly assumes categorical variables are ordinal.

**"Built-in" Methods**   These are methods for treating missing values that do not require external models, and hence are "built-in" into the decision tree structure. In fact, they consist of slight modifications to the inference procedure.

> **Surrogate splits** [3, 20]: During training, once the best split is defined, one ranks alternative splits on the number of instances that they send to the same branch as the best split. At test time, if the split variable is not observed, one tries the surrogate splits in order (starting with that which most resembles the best split). If none of the surrogate split variables is available, the instance is sent to the branch with the highest number of data points at training time. Surrogate splits have two notable drawbacks: (i) their performance is heavily dependent on the correlation between variables; (ii) they require storing every possible split to be guaranteed to work for all missing-value configurations, which is rather computational intensive, especially for large ensembles.
>
> **Friedman method** [6, 16]: Whenever a split variable is not observed, one follows both branches of the tree. That means any instance with missing value is mapped to multiple leaves, and the final prediction is given by the majority class across the sum of the counts of all these leaves. If $C^{\mathcal{A}}(j)$ gives the number of training instances of class $j$ in cell $\mathcal{A}$, we can write Friedman's methods as

$$f(\mathbf{x}) = \underset{j \in \{1, \ldots, K\}}{\operatorname{argmax}} \sum_{\mathcal{A} \in \boldsymbol{\mathcal{A}}} \mathbb{1}(\mathbf{x} \in \mathcal{A}) C^{\mathcal{A}}(j),$$

$$C^{\mathcal{A}}(j) = \sum_{i=1}^{n} \mathbb{1}(\mathbf{x}_i \in \mathcal{A}) \mathbb{1}(y_i = j),$$

where $i$ runs through the $n$ training instances $(\mathbf{x}_i, y_i)$, and $j$ runs through the $K$ possible classes. Note that Friedman's method can be seen as a simplified version of GeFs where the density over explanatory variables is constant and the same in every leaf.

**Imputation methods**    It is not surprising that most of the work on handling missing data in decision trees and random forests rely on data imputation [17]. That is, another or multiple other models are used to predict the missing values before feeding the data to the tree-based classifier. In the experiments we compare two different types of imputation methods:

**Mean** Missing values are imputed with the mean for continuous variables or the most frequent observation for categorical variables.

**KNN** Similar to the simple method above but the means or most frequent values are taken over the $K$-nearest neighbours. We use a standard K-nearest neighbour implementation from `scikit-learn` [15] with K=7. However, the distance function is updated to better accommodate mixed data types. Following, Huang et al. [10], we define the distance measure as

$$d(\mathbf{x}_a, \mathbf{x}_b) = \gamma \sum_{i=0}^{m_0} w_i \delta(\mathbf{x}_a[i], \mathbf{x}_b[i]) + \sum_{i=m_0}^{m_1} w_i \sqrt{(\mathbf{x}_a[i] - \mathbf{x}_b[i])^2},$$

where $\gamma$ is a parameter representing the relative importance of categorical and numerical features, $w_i$ is the weight of feature $i$, and, without loss of generality, we assume features are ordered so that the first $m_0$ variables are categorical. The $\delta$ function is simply the Hamming distance: $\delta(\mathbf{x}_a[i], \mathbf{x}_b[i]) = 1$ if $\mathbf{x}_a[i] \neq \mathbf{x}_b[i]$, and $\delta(\mathbf{x}_a[i], \mathbf{x}_b[i]) = 0$ otherwise. As we have no reason to favour any feature or feature type, we set both $\gamma$ and every $w_i$ to one.

**MissForest** [19]: For each variable $X_i \in \mathbf{X}$, one learns a Random Forest (classifier/regressor) that is used to predict unobserved values of $X_i$ given the other variables $\mathbf{X} \setminus X_i$. As more than one variable might be unobserved, MissForest starts by imputing missing values with the mean (or mode) and then iteratively updates its initial guess using the Random Forest predictors. The original MissForest algorithm proposed in [19] also updates the Random Forest predictors at every iteration. However, in our experiments that would allow MissForest to exploit test data information, which could compromise the results. Therefore, we fit the Random Forest predictors in the training data only and keep them fixed at test time. Note that the algorithm remains iterative, since the imputed values are still fed to the predictors in the next iteration. We use a standard Python implementation of MissForests from `missingpy`—adapted to accommodate the changes mentioned above—which relies on the `scikit-learn` implementation of Random Forests.

**Vanilla GeFs**    What we call *vanilla* GeF, or simply GeF, is a model where the distribution at the leaves is given by a fully factorised model, that is, for each leaf $v$, $p_v(\mathbf{x}, y) = p_v(x_1)p_v(x_2)\ldots p_v(x_m)p_v(y)$. This is probably the simplest model that one can fit at the leaves and is clearly *class-factorised*. Therefore, vanilla GeFs preserve full backward-compatibility with the original RF, yielding the exact same prediction function for complete data.

**GeF with LearnSPN**    For GeF(LearnSPN) and GeF$^+$(LearnSPN), the LearnSPN algorithm [8] is run only at leaves with more than 30 samples, and smaller leaves are modelled by a fully factorised model as in vanilla GeFs. That saves computational time with little performance impact, as the model derived from LearnSPN with few samples would be similarly simplistic. We run the LearnSPN algorithm as follows: sum nodes split the samples via K-means clustering with K=2, and product nodes split the variables with an independence threshold of 0.001 (pair of variables for which the independence test yields a p-value lower than the threshold are considered independent). We do not force independence between the class $Y$ and input variables $\mathbf{X}$ in LearnSPN, which explains why, in contrast to GeF, GeF(LearnSPN) does not necessarily yield the same predictions as the original Random Forest.

**LearnSPN**    Similarly, we also learn a Probabilistic Circuit by applying the LearnSPN algorithm [8] to the entire dataset. The hyperparameters for this experiment are the same as for GeFs with LearnSPN, but we use a variant of LearnSPN that yields class-selective PCs, which have been shown to outperform standard LearnSPN in classification tasks [4].

## C.1 (Banknote) Authentication [5]

Dataset details

| n | $m_0$ | $m_1$ | $|\mathcal{Y}|$ | %Maj |
|---|---|---|---|---|
| 1372 | 0 | 4 | 2 | 55.54 |

Table 1: Accuracy per percent of missing values at test time with 95% confidence intervals.

| (%) | Surrogate | Friedman | Mean | KNN | MissForest |
|---|---|---|---|---|---|
| 0 | 98.99 ± .12 | 98.99 ± .12 | 98.99 ± .12 | 98.99 ± .12 | 98.99 ± .12 |
| 10 | 95.93 ± .45 | 95.44 ± .37 | 94.22 ± .59 | **97.52** ± .17 | 97.33 ± .19 |
| 20 | 92.51 ± .57 | 91.44 ± .45 | 89.19 ± .76 | **95.08** ± .42 | **94.74** ± .29 |
| 30 | 88.65 ± .74 | 87.13 ± .69 | 84.47 ± .79 | **91.98** ± .55 | 90.74 ± .37 |
| 40 | 84.46 ± .74 | 82.54 ± .55 | 79.59 ± .69 | **87.92** ± .67 | 86.21 ± .55 |
| 50 | 80.04 ± .84 | 78.35 ± .48 | 74.77 ± .84 | **83.18** ± 1.02 | 80.68 ± .69 |
| 60 | 75.05 ± 1.06 | 73.43 ± .92 | 70.36 ± 1.11 | **78.08** ± 1.18 | 74.74 ± 1.04 |
| 70 | 69.96 ± 1.03 | 69.36 ± 1.02 | 66.07 ± .85 | **72.54** ± 1.71 | 68.57 ± 1.85 |
| 80 | 65.04 ± .55 | 65.16 ± 1.00 | 62.6 ± .65 | **67.25** ± 1.16 | 62.78 ± 2.05 |
| 90 | 58.99 ± .68 | 59.89 ± .92 | 58.65 ± .71 | **61.26** ± .90 | 54.27 ± 2.05 |

| (%) | LearnSPN | GeF | GeF(LSPN) | $\text{GeF}^+$ | $\text{GeF}^+$(LSPN) |
|---|---|---|---|---|---|
| 0 | 91.12 ± .52 | 98.99 ± .12 | 98.99 ± .12 | 99.07 ± .08 | **99.45** ± .08 |
| 10 | 88.15 ± .65 | 96.87 ± .25 | 96.47 ± .31 | 96.81 ± .27 | 96.74 ± .29 |
| 20 | 84.99 ± .81 | 93.97 ± .34 | 93.44 ± .35 | 93.91 ± .45 | 93.46 ± .49 |
| 30 | 81.65 ± .94 | 90.33 ± .65 | 89.66 ± .64 | 90.3 ± .70 | 89.52 ± .71 |
| 40 | 78.05 ± .79 | 85.99 ± .72 | 85.31 ± .55 | 85.96 ± .59 | 85.07 ± .65 |
| 50 | 74.67 ± .79 | 81.66 ± .83 | 80.94 ± .68 | 81.52 ± .72 | 80.64 ± .73 |
| 60 | 70.97 ± 1.03 | 76.32 ± 1.00 | 75.58 ± .88 | 76.12 ± .94 | 75.22 ± .79 |
| 70 | 67.4 ± 1.02 | **71.54** ± 1.27 | **70.84** ± 1.04 | **71.3** ± 1.13 | 70.47 ± .95 |
| 80 | 64.34 ± .84 | **66.74** ± .90 | **66.28** ± .77 | **66.53** ± .80 | 65.95 ± .67 |
| 90 | 59.69 ± .79 | **60.74** ± 1.06 | **60.52** ± 1.01 | **60.58** ± 1.03 | 60.23 ± .90 |

Figure 2: Accuracy against proportion of missing values. The same plot is repeated ten times, each time highlighting a different method and its 95% confidence interval.

## C.2 Bank Marketing [14]

Dataset details

| n | $m_0$ | $m_1$ | $\lvert \mathcal{Y} \rvert$ | %Maj |
|---|---|---|---|---|
| 41188 | 11 | 9 | 2 | 88.73 |

Table 2: Accuracy per percent of missing values at test time with 95% confidence intervals.

| (%) | Surrogate | Friedman | Mean | KNN | MissForest |
|---|---|---|---|---|---|
| 0 | **91.39** ± .19 | **91.39** ± .19 | **91.39** ± .19 | **91.39** ± .19 | **91.39** ± .19 |
| 10 | **91.1** ± .21 | **91.08** ± .15 | 90.94 ± .14 | **91.13** ± .17 | **91.13** ± .16 |
| 20 | 90.74 ± .20 | 90.7 ± .14 | 90.48 ± .13 | **90.89** ± .15 | 90.79 ± .17 |
| 30 | 90.42 ± .18 | 90.3 ± .15 | 90.09 ± .11 | **90.64** ± .14 | 90.4 ± .19 |
| 40 | 90.09 ± .18 | 89.87 ± .11 | 89.68 ± .12 | 90.33 ± .16 | 89.99 ± .21 |
| 50 | 89.78 ± .19 | 89.47 ± .07 | 89.4 ± .10 | 90.02 ± .25 | 89.54 ± .21 |
| 60 | 89.46 ± .23 | 89.13 ± .08 | 89.17 ± .16 | 89.74 ± .26 | 89.09 ± .24 |
| 70 | 89.11 ± .20 | 88.89 ± .07 | 88.96 ± .08 | 89.33 ± .28 | 88.58 ± .32 |
| 80 | 88.84 ± .22 | 88.79 ± .02 | 88.85 ± .05 | 88.99 ± .13 | 88.37 ± .36 |
| 90 | 88.74 ± .12 | 88.74 ± .01 | 88.78 ± .04 | 88.7 ± .16 | 88.23 ± .44 |

| (%) | LearnSPN | GeF | GeF(LSPN) | GeF$^+$ | GeF$^+$(LSPN) |
|---|---|---|---|---|---|
| 0 | 89.14 ± .09 | **91.39** ± .19 | **91.41** ± .20 | 90.23 ± .32 | 89.77 ± .36 |
| 10 | 89.21 ± .06 | **91.19** ± .17 | **91.23** ± .20 | 89.67 ± .30 | 88.22 ± .32 |
| 20 | 89.3 ± .05 | **91.03** ± .15 | **91.02** ± .15 | 89.36 ± .26 | 87.24 ± .32 |
| 30 | 89.39 ± .05 | **90.79** ± .21 | **90.77** ± .21 | 89.25 ± .33 | 86.79 ± .43 |
| 40 | 89.38 ± .06 | **90.52** ± .17 | **90.46** ± .19 | 89.33 ± .37 | 86.93 ± .49 |
| 50 | 89.38 ± .05 | **90.31** ± .18 | **90.26** ± .13 | 89.45 ± .25 | 87.36 ± .42 |
| 60 | 89.35 ± .05 | **90.03** ± .18 | **90.02** ± .17 | 89.59 ± .20 | 88.1 ± .31 |
| 70 | 89.27 ± .06 | **89.69** ± .20 | **89.67** ± .22 | 89.48 ± .21 | 88.65 ± .38 |
| 80 | 89.16 ± .04 | **89.37** ± .18 | **89.37** ± .19 | **89.33** ± .22 | 89.01 ± .26 |
| 90 | **88.99** ± .02 | **89.05** ± .09 | **89.05** ± .09 | **89.04** ± .09 | **88.99** ± .09 |

Figure 3: Accuracy against proportion of missing values. The same plot is repeated ten times, each time highlighting a different method and its 95% confidence interval.

## C.3 Breast Cancer (WDBC) [3]

Dataset details

| n | $m_0$ | $m_1$ | $|\mathcal{Y}|$ | %Maj |
|---|---|---|---|---|
| 569 | 0 | 30 | 2 | 62.74 |

Table 3: Accuracy per percent of missing values at test time with 95% confidence intervals.

| (%) | Surrogate | Friedman | Mean | KNN | MissForest |
|---|---|---|---|---|---|
| 0 | $95.69 \pm .30$ | $95.69 \pm .30$ | $95.69 \pm .30$ | $95.69 \pm .30$ | $95.69 \pm .30$ |
| 10 | $95.47 \pm .33$ | $95.39 \pm .37$ | $95.36 \pm .30$ | $95.61 \pm .14$ | $95.55 \pm .35$ |
| 20 | $95.24 \pm .25$ | $95.17 \pm .40$ | $95.16 \pm .32$ | $95.52 \pm .26$ | $95.27 \pm .24$ |
| 30 | $94.96 \pm .36$ | $94.96 \pm .35$ | $94.92 \pm .58$ | $95.59 \pm .41$ | $94.92 \pm .36$ |
| 40 | $94.76 \pm .57$ | $94.5 \pm .43$ | $94.25 \pm .56$ | $95.34 \pm .45$ | $94.74 \pm .38$ |
| 50 | $94.2 \pm .45$ | $94.13 \pm .31$ | $93.32 \pm .62$ | $94.81 \pm .42$ | $93.88 \pm .47$ |
| 60 | $93.21 \pm .40$ | $92.99 \pm .41$ | $91.66 \pm .78$ | $\mathbf{94.8} \pm .42$ | $92.9 \pm .28$ |
| 70 | $90.85 \pm .55$ | $90.74 \pm .71$ | $88.88 \pm 1.08$ | $\mathbf{93.78} \pm .26$ | $91.0 \pm .96$ |
| 80 | $84.89 \pm .98$ | $85.98 \pm 1.27$ | $83.67 \pm 1.55$ | $\underline{\mathbf{92.14}} \pm .68$ | $86.03 \pm 1.67$ |
| 90 | $72.39 \pm .71$ | $75.89 \pm 1.12$ | $73.48 \pm 1.75$ | $\underline{\mathbf{84.97}} \pm .97$ | $76.71 \pm 1.38$ |

| (%) | LearnSPN | GeF | GeF(LSPN) | GeF$^+$ | GeF$^+$(LSPN) |
|---|---|---|---|---|---|
| 0 | $95.49 \pm .24$ | $95.69 \pm .30$ | $95.75 \pm .28$ | $\mathbf{96.61} \pm .32$ | $\underline{\mathbf{96.66}} \pm .37$ |
| 10 | $95.47 \pm .33$ | $95.64 \pm .39$ | $95.87 \pm .20$ | $\mathbf{96.4} \pm .22$ | $\underline{\mathbf{96.54}} \pm .32$ |
| 20 | $95.15 \pm .28$ | $95.67 \pm .24$ | $96.17 \pm .27$ | $96.22 \pm .23$ | $\underline{\mathbf{96.66}} \pm .29$ |
| 30 | $94.85 \pm .29$ | $95.64 \pm .47$ | $\mathbf{96.26} \pm .42$ | $95.82 \pm .21$ | $\underline{\mathbf{96.27}} \pm .35$ |
| 40 | $94.52 \pm .26$ | $95.37 \pm .41$ | $\underline{\mathbf{95.99}} \pm .53$ | $95.2 \pm .27$ | $\mathbf{95.92} \pm .40$ |
| 50 | $93.99 \pm .48$ | $94.81 \pm .44$ | $\underline{\mathbf{95.57}} \pm .45$ | $94.71 \pm .41$ | $\mathbf{95.36} \pm .31$ |
| 60 | $93.39 \pm .37$ | $94.09 \pm .53$ | $\underline{\mathbf{95.18}} \pm .48$ | $94.04 \pm .43$ | $\mathbf{94.9} \pm .40$ |
| 70 | $91.94 \pm .25$ | $92.72 \pm .40$ | $\underline{\mathbf{93.86}} \pm .42$ | $92.36 \pm .51$ | $\mathbf{93.5} \pm .26$ |
| 80 | $89.25 \pm .71$ | $90.72 \pm .50$ | $91.37 \pm .63$ | $90.07 \pm .44$ | $90.81 \pm .54$ |
| 90 | $82.46 \pm .72$ | $\mathbf{84.29} \pm .88$ | $\underline{\mathbf{84.62}} \pm .81$ | $83.67 \pm .65$ | $83.87 \pm .74$ |

Figure 4: Accuracy against proportion of missing values. The same plot is repeated ten times, each time highlighting a different method and its 95% confidence interval.

[3]This breast cancer domain was obtained from the University Medical Centre, Institute of Oncology, Ljubljana, Yugoslavia. Thanks go to M. Zwitter and M. Soklic for providing the data.

## C.4 Contraceptive Method Choice (CMC) [5]

Dataset details

| n | $m_0$ | $m_1$ | $|\mathcal{Y}|$ | %Maj |
|---|---|---|---|---|
| 1473 | 8 | 1 | 3 | 42.7 |

Table 4: Accuracy per percent of missing values at test time with 95% confidence intervals.

| (%) | Surrogate | Friedman | Mean | KNN | MissForest |
|---|---|---|---|---|---|
| 0 | **53.27** ± .51 | **53.27** ± .51 | **53.27** ± .51 | **53.27** ± .51 | **53.27** ± .51 |
| 10 | **51.7** ± .55 | **52.19** ± .65 | 51.18 ± .58 | **51.67** ± .51 | **51.64** ± .46 |
| 20 | 50.35 ± .42 | **50.73** ± .50 | 49.28 ± .90 | 50.1 ± .51 | 50.14 ± .59 |
| 30 | 48.7 ± .79 | **49.8** ± .38 | 47.67 ± .86 | 48.38 ± .58 | 48.28 ± .90 |
| 40 | 46.08 ± .77 | **48.3** ± .42 | 45.88 ± .67 | 46.79 ± .77 | 45.62 ± .78 |
| 50 | 43.73 ± 1.03 | **47.31** ± .40 | 44.3 ± .79 | 45.56 ± .59 | 43.57 ± .63 |
| 60 | 41.63 ± 1.11 | 46.14 ± .44 | 42.78 ± 1.26 | 44.3 ± .83 | 41.32 ± .83 |
| 70 | 39.64 ± 1.27 | 45.09 ± .33 | 41.79 ± 1.00 | 42.3 ± .69 | 39.04 ± .84 |
| 80 | 37.43 ± 1.44 | **43.97** ± .43 | 40.85 ± 1.07 | 41.28 ± .51 | 37.17 ± 1.08 |
| 90 | 34.97 ± 1.81 | **43.33** ± .31 | 40.24 ± 1.10 | 39.64 ± .98 | 33.56 ± 1.78 |

| (%) | LearnSPN | GeF | GeF(LSPN) | GeF$^+$ | GeF$^+$(LSPN) |
|---|---|---|---|---|---|
| 0 | 49.66 ± 1.01 | **53.27** ± .51 | 53.24 ± .56 | 50.9 ± .57 | 50.71 ± .56 |
| 10 | 49.48 ± 1.03 | **52.02** ± .59 | 52.04 ± .54 | 50.03 ± .64 | 49.9 ± .66 |
| 20 | 49.23 ± .86 | **51.05** ± .65 | **51.08** ± .59 | 49.8 ± .74 | 49.72 ± .78 |
| 30 | 48.4 ± 1.01 | **49.96** ± 1.03 | **50.08** ± 1.05 | 48.81 ± .91 | 48.72 ± .90 |
| 40 | 47.81 ± 1.05 | **48.71** ± .85 | **48.84** ± .86 | 47.84 ± .85 | 47.85 ± .94 |
| 50 | 46.84 ± 1.06 | **47.92** ± .63 | **47.91** ± .66 | **47.47** ± .65 | **47.42** ± .68 |
| 60 | 46.47 ± .97 | **47.09** ± .47 | **47.01** ± .40 | **46.75** ± .51 | **46.71** ± .41 |
| 70 | **45.53** ± .99 | **45.94** ± .57 | 45.93 ± .63 | 45.67 ± .67 | 45.67 ± .69 |
| 80 | **44.07** ± .70 | **44.28** ± .50 | **44.28** ± .50 | 44.2 ± .51 | 44.22 ± .47 |
| 90 | **43.74** ± .55 | **43.67** ± .69 | **43.76** ± .73 | 43.61 ± .69 | 43.72 ± .73 |

Figure 5: Accuracy against proportion of missing values. The same plot is repeated ten times, each time highlighting a different method and its 95% confidence interval.

## C.5 Credit-g [5]

Dataset details

| n | $m_0$ | $m_1$ | $|\mathcal{Y}|$ | %Maj |
|---|---|---|---|---|
| 1000 | 13 | 7 | 2 | 70.0 |

Table 5: Accuracy per percent of missing values at test time with 95% confidence intervals.

| (%) | Surrogate | Friedman | Mean | KNN | MissForest |
|---|---|---|---|---|---|
| 0 | **75.75** ± .44 | **75.75** ± .44 | **75.75** ± .44 | **75.75** ± .44 | **75.75** ± .44 |
| 10 | 74.52 ± .45 | 74.57 ± .40 | 74.56 ± .47 | 74.71 ± .41 | 74.55 ± .51 |
| 20 | 73.33 ± .32 | 73.86 ± .49 | 73.95 ± .30 | 73.8 ± .44 | 73.76 ± .68 |
| 30 | 71.97 ± .32 | 72.42 ± .31 | 73.01 ± .64 | 73.06 ± .65 | 73.03 ± .78 |
| 40 | 71.09 ± .27 | 71.72 ± .43 | 72.21 ± .51 | 72.27 ± .92 | 72.34 ± .82 |
| 50 | 70.58 ± .28 | 71.08 ± .30 | 71.43 ± .58 | 71.48 ± 1.02 | 70.94 ± .80 |
| 60 | 70.42 ± .27 | 70.69 ± .33 | 70.92 ± .44 | 70.29 ± .79 | 69.82 ± .78 |
| 70 | 70.16 ± .10 | 70.31 ± .16 | 70.57 ± .46 | 69.09 ± 1.02 | 69.36 ± .82 |
| 80 | 70.07 ± .10 | 70.09 ± .08 | 70.32 ± .31 | 67.45 ± .82 | 68.76 ± .90 |
| 90 | 69.99 ± .02 | 69.99 ± .02 | **70.01** ± .11 | 66.8 ± 1.37 | 68.53 ± 1.15 |

| (%) | LearnSPN | GeF | GeF(LSPN) | GeF$^+$ | GeF$^+$(LSPN) |
|---|---|---|---|---|---|
| 0 | 73.41 ± .87 | **75.75** ± .44 | **75.76** ± .43 | 74.72 ± .63 | 74.5 ± .72 |
| 10 | 72.7 ± .66 | **74.98** ± .38 | **75.07** ± .32 | 73.97 ± .71 | 73.73 ± .64 |
| 20 | 72.32 ± .77 | **74.68** ± .45 | **74.82** ± .45 | 73.7 ± .76 | 73.36 ± .76 |
| 30 | 72.14 ± .65 | **73.81** ± .38 | **73.97** ± .36 | 72.74 ± .70 | 72.41 ± .63 |
| 40 | 71.96 ± .59 | **73.47** ± .45 | **73.58** ± .38 | 72.32 ± .44 | 71.92 ± .51 |
| 50 | 71.25 ± .46 | **72.82** ± .57 | **72.97** ± .60 | 71.74 ± .71 | 71.48 ± .68 |
| 60 | 70.83 ± .87 | **72.14** ± .34 | **72.02** ± .48 | 71.29 ± .59 | 71.21 ± .59 |
| 70 | 70.59 ± .60 | **71.24** ± .42 | **71.28** ± .40 | **70.94** ± .51 | **70.94** ± .47 |
| 80 | **70.43** ± .38 | **70.63** ± .32 | **70.77** ± .34 | **70.61** ± .32 | **70.61** ± .41 |
| 90 | **70.37** ± .36 | **70.33** ± .33 | **70.34** ± .40 | **70.25** ± .41 | **70.27** ± .41 |

Figure 6: Accuracy against proportion of missing values. The same plot is repeated ten times, each time highlighting a different method and its 95% confidence interval.

## C.6 Diabetes [5]

Dataset details

| n | $m_0$ | $m_1$ | $|\mathcal{Y}|$ | %Maj |
|---|---|---|---|---|
| 768 | 0 | 8 | 2 | 65.1 |

Table 6: Accuracy per percent of missing values at test time with 95% confidence intervals.

| (%) | Surrogate | Friedman | Mean | KNN | MissForest |
|---|---|---|---|---|---|
| 0 | **75.88** ± .46 | **75.88** ± .46 | **75.88** ± .46 | **75.88** ± .46 | **75.88** ± .46 |
| 10 | **75.06** ± .71 | **75.23** ± .62 | 74.53 ± .78 | **75.1** ± .92 | **74.87** ± .89 |
| 20 | **73.96** ± .81 | **73.91** ± .81 | 72.99 ± 1.00 | 73.45 ± .83 | **73.61** ± .65 |
| 30 | 72.97 ± .73 | **73.35** ± .70 | 71.67 ± .84 | 72.4 ± .75 | 72.46 ± .92 |
| 40 | **72.45** ± .64 | **72.48** ± .98 | 69.99 ± .83 | 71.41 ± .73 | 71.85 ± .61 |
| 50 | **71.54** ± .70 | **71.76** ± .79 | 68.11 ± .62 | 69.95 ± .80 | 70.6 ± .77 |
| 60 | **70.7** ± .52 | **70.78** ± .94 | 66.89 ± .81 | 67.93 ± 1.02 | 68.94 ± .80 |
| 70 | 68.98 ± .73 | **69.31** ± .82 | 65.51 ± .56 | 65.79 ± 1.10 | 66.7 ± 1.25 |
| 80 | **67.76** ± .55 | 67.67 ± .47 | 64.65 ± .78 | 64.07 ± .96 | 64.9 ± .66 |
| 90 | **66.6** ± .69 | **66.45** ± .55 | 64.05 ± 1.08 | 63.18 ± 1.35 | 63.05 ± 2.43 |

| (%) | LearnSPN | GeF | GeF(LSPN) | $\text{GeF}^+$ | $\text{GeF}^+$(LSPN) |
|---|---|---|---|---|---|
| 0 | 75.08 ± .62 | **75.88** ± .46 | **75.88** ± .46 | 75.39 ± 1.01 | 74.47 ± 1.05 |
| 10 | **74.65** ± .59 | **75.31** ± .67 | **75.31** ± .70 | **74.83** ± .99 | 74.25 ± 1.09 |
| 20 | 73.85 ± .61 | **74.21** ± .65 | **74.18** ± .75 | **74.18** ± .98 | 73.53 ± 1.05 |
| 30 | 72.86 ± .63 | **73.93** ± .63 | **73.83** ± .72 | 73.28 ± .87 | 72.81 ± 1.01 |
| 40 | 72.03 ± 1.00 | **73.07** ± .80 | **72.97** ± .91 | **72.57** ± .85 | 72.18 ± .78 |
| 50 | **71.59** ± .91 | **72.06** ± .68 | **72.16** ± .68 | **71.5** ± .80 | 71.28 ± .78 |
| 60 | **70.66** ± .91 | **71.46** ± .83 | **71.3** ± .72 | **71.24** ± .70 | **70.95** ± .78 |
| 70 | **69.14** ± .83 | **69.91** ± .90 | **69.82** ± .80 | **69.84** ± .92 | **69.5** ± .85 |
| 80 | **67.82** ± .79 | **68.46** ± .69 | **68.39** ± .54 | **68.48** ± .80 | **68.42** ± .70 |
| 90 | **66.46** ± .59 | **66.72** ± .67 | **66.9** ± .59 | **66.75** ± .68 | **66.84** ± .59 |

Figure 7: Accuracy against proportion of missing values. The same plot is repeated ten times, each time highlighting a different method and its 95% confidence interval.

## C.7 DNA (Primate splice-junction gene sequences) [5]

Dataset details

| n | $m_0$ | $m_1$ | $|\mathcal{Y}|$ | %Maj |
|---|---|---|---|---|
| 3186 | 180 | 0 | 3 | 51.91 |

This is the same dataset as Splice, but here the categorical variables were one-hot encoded.

Table 7: Accuracy per percent of missing values at test time with 95% confidence intervals.

| (%) | Surrogate | Friedman | Mean | KNN | MissForest |
|---|---|---|---|---|---|
| 0 | **95.34** ± .13 | **95.34** ± .13 | **95.34** ± .13 | **95.34** ± .13 | **95.34** ± .13 |
| 10 | **94.32** ± .24 | 91.03 ± .23 | 92.16 ± .27 | 93.89 ± .16 | **94.46** ± .17 |
| 20 | 92.76 ± .33 | 84.68 ± .29 | 88.28 ± .52 | 91.96 ± .33 | 93.02 ± .22 |
| 30 | 90.53 ± .31 | 77.23 ± .36 | 83.91 ± .43 | 89.31 ± .33 | 90.76 ± .34 |
| 40 | 87.62 ± .37 | 70.13 ± .30 | 79.0 ± .46 | 86.2 ± .37 | 87.32 ± .38 |
| 50 | 83.34 ± .23 | 63.85 ± .42 | 72.73 ± .29 | 81.87 ± .35 | 82.26 ± .54 |
| 60 | 77.11 ± .45 | 58.75 ± .35 | 65.53 ± .36 | 76.8 ± .44 | 75.05 ± .50 |
| 70 | 68.38 ± .67 | 55.2 ± .23 | 56.69 ± .54 | 71.18 ± .62 | 65.34 ± .71 |
| 80 | 55.84 ± .82 | 53.07 ± .17 | 46.46 ± .99 | 64.37 ± .43 | 54.12 ± .75 |
| 90 | 40.14 ± .75 | 52.06 ± .06 | 35.38 ± 1.33 | 56.82 ± .95 | 41.05 ± .51 |

| (%) | LearnSPN | GeF | GeF(LSPN) | GeF$^+$ | GeF$^+$(LSPN) |
|---|---|---|---|---|---|
| 0 | 92.82 ± .43 | **95.34** ± .13 | **95.33** ± .13 | **95.3** ± .07 | 93.15 ± .14 |
| 10 | 92.23 ± .28 | 92.28 ± .27 | 91.31 ± .18 | **94.32** ± .21 | 92.1 ± .21 |
| 20 | 91.17 ± .40 | 89.6 ± .28 | 87.14 ± .28 | **93.3** ± .23 | 91.15 ± .24 |
| 30 | 89.62 ± .36 | 87.42 ± .19 | 82.99 ± .25 | **92.04** ± .32 | 90.21 ± .23 |
| 40 | 88.02 ± .31 | 85.83 ± .26 | 80.38 ± .33 | **90.41** ± .21 | 89.0 ± .18 |
| 50 | 85.63 ± .33 | 84.45 ± .27 | 78.87 ± .34 | **88.32** ± .39 | 87.17 ± .32 |
| 60 | 82.14 ± .36 | 82.82 ± .37 | 78.43 ± .32 | **85.08** ± .36 | 84.44 ± .25 |
| 70 | 77.23 ± .56 | 80.15 ± .33 | 77.97 ± .35 | **81.11** ± .42 | **80.87** ± .33 |
| 80 | 70.85 ± .66 | 74.82 ± .41 | 74.58 ± .41 | **75.27** ± .43 | 75.22 ± .47 |
| 90 | 61.27 ± .49 | 64.09 ± .55 | 64.6 ± .54 | **65.06** ± .60 | **65.19** ± .55 |

Figure 8: Accuracy against proportion of missing values. The same plot is repeated ten times, each time highlighting a different method and its 95% confidence interval.

## C.8 Dresses-sales [5]

### Dataset details

| n | $m_0$ | $m_1$ | $|\mathcal{Y}|$ | %Maj |
|---|---|---|---|---|
| 500 | 12 | 0 | 2 | 58.0 |

Table 8: Accuracy per percent of missing values at test time with 95% confidence intervals.

| (%) | Surrogate | Friedman | Mean | KNN | MissForest |
|---|---|---|---|---|---|
| 0 | $56.06 \pm .86$ | $56.06 \pm .86$ | $56.06 \pm .86$ | $56.06 \pm .86$ | $56.06 \pm .86$ |
| 10 | $50.56 \pm 1.32$ | $56.46 \pm 1.36$ | $57.14 \pm 1.04$ | $56.4 \pm 1.29$ | $56.32 \pm 1.25$ |
| 20 | $47.22 \pm 1.82$ | $56.1 \pm 1.27$ | $\mathbf{58.16} \pm 1.33$ | $56.92 \pm 1.46$ | $56.44 \pm 1.65$ |
| 30 | $45.48 \pm 1.57$ | $55.8 \pm 1.21$ | $\mathbf{58.18} \pm .85$ | $56.62 \pm 1.57$ | $55.68 \pm .95$ |
| 40 | $43.78 \pm 1.15$ | $56.22 \pm 1.07$ | $\mathbf{57.86} \pm 1.35$ | $56.26 \pm 2.10$ | $55.94 \pm 1.39$ |
| 50 | $43.12 \pm .80$ | $57.24 \pm 1.22$ | $\mathbf{57.98} \pm 1.04$ | $55.92 \pm 1.21$ | $55.04 \pm 1.41$ |
| 60 | $42.3 \pm .55$ | $\mathbf{57.7} \pm 1.33$ | $57.96 \pm .81$ | $56.08 \pm 1.00$ | $55.42 \pm 1.56$ |
| 70 | $42.0 \pm .38$ | $\underline{\mathbf{58.28}} \pm .96$ | $57.88 \pm 1.16$ | $56.86 \pm 1.46$ | $55.34 \pm 1.92$ |
| 80 | $42.04 \pm .16$ | $\underline{\mathbf{58.34}} \pm .98$ | $58.0 \pm .58$ | $57.0 \pm 1.89$ | $55.04 \pm 1.17$ |
| 90 | $42.04 \pm .09$ | $\underline{\mathbf{58.64}} \pm .78$ | $57.82 \pm .64$ | $57.28 \pm 1.50$ | $54.92 \pm 2.25$ |

| (%) | LearnSPN | GeF | GeF(LSPN) | GeF$^+$ | GeF$^+$(LSPN) |
|---|---|---|---|---|---|
| 0 | $\underline{\mathbf{57.46}} \pm .82$ | $56.06 \pm .86$ | $56.08 \pm .87$ | $56.42 \pm 1.01$ | $56.38 \pm 1.01$ |
| 10 | $\underline{\mathbf{58.14}} \pm .78$ | $56.62 \pm 1.55$ | $56.58 \pm 1.57$ | $56.36 \pm .65$ | $56.22 \pm .69$ |
| 20 | $\underline{\mathbf{58.64}} \pm 1.19$ | $57.04 \pm 1.53$ | $57.02 \pm 1.53$ | $56.38 \pm .94$ | $56.3 \pm 1.08$ |
| 30 | $\underline{\mathbf{58.24}} \pm 1.07$ | $57.12 \pm 1.11$ | $57.14 \pm 1.07$ | $55.5 \pm 1.41$ | $55.52 \pm 1.47$ |
| 40 | $\underline{\mathbf{58.08}} \pm .89$ | $\mathbf{57.36} \pm .79$ | $\mathbf{57.38} \pm .82$ | $55.08 \pm 1.24$ | $55.16 \pm 1.23$ |
| 50 | $\underline{\mathbf{58.3}} \pm .96$ | $56.78 \pm .93$ | $56.8 \pm .94$ | $54.86 \pm 1.09$ | $54.9 \pm 1.05$ |
| 60 | $\underline{\mathbf{58.2}} \pm 1.05$ | $56.24 \pm 1.29$ | $56.24 \pm 1.32$ | $55.52 \pm 1.36$ | $55.56 \pm 1.34$ |
| 70 | $\mathbf{57.7} \pm .78$ | $57.0 \pm 1.32$ | $57.02 \pm 1.33$ | $56.42 \pm 1.36$ | $56.46 \pm 1.37$ |
| 80 | $\mathbf{57.8} \pm 1.05$ | $57.04 \pm 1.11$ | $57.02 \pm 1.10$ | $56.46 \pm 1.01$ | $56.44 \pm 1.03$ |
| 90 | $\mathbf{58.22} \pm 1.13$ | $57.68 \pm 1.32$ | $57.68 \pm 1.32$ | $57.6 \pm 1.18$ | $57.6 \pm 1.18$ |

Figure 9: Accuracy against proportion of missing values. The same plot is repeated ten times, each time highlighting a different method and its 95% confidence interval.

## C.9 Electricity [7]

Dataset details

| n | $m_0$ | $m_1$ | $|\mathcal{Y}|$ | %Maj |
|---|---|---|---|---|
| 45312 | 1 | 7 | 2 | 57.55 |

Table 9: Accuracy per percent of missing values at test time with 95% confidence intervals.

| (%) | Surrogate | Friedman | Mean | KNN | MissForest |
|---|---|---|---|---|---|
| 0 | **91.22** ± .08 | **91.22** ± .08 | **91.22** ± .08 | **91.22** ± .08 | **91.22** ± .08 |
| 10 | 87.49 ± .09 | 85.73 ± .09 | 84.91 ± .10 | 87.78 ± .08 | 88.54 ± .07 |
| 20 | 83.62 ± .06 | 81.27 ± .07 | 78.9 ± .19 | 84.24 ± .11 | 85.2 ± .08 |
| 30 | 79.79 ± .09 | 77.47 ± .10 | 73.24 ± .19 | 80.55 ± .11 | 81.21 ± .10 |
| 40 | 75.99 ± .08 | 74.37 ± .14 | 67.82 ± .25 | 76.74 ± .16 | 76.69 ± .13 |
| 50 | 72.15 ± .09 | 71.58 ± .16 | 62.74 ± .35 | 72.81 ± .11 | 71.78 ± .13 |
| 60 | 68.32 ± .13 | 68.85 ± .18 | 58.05 ± .44 | 68.58 ± .09 | 66.49 ± .31 |
| 70 | 64.25 ± .25 | 66.12 ± .18 | 53.69 ± .51 | 64.04 ± .14 | 61.58 ± .64 |
| 80 | 59.93 ± .50 | 63.38 ± .21 | 49.92 ± .64 | 58.7 ± .27 | 55.97 ± .52 |
| 90 | 54.84 ± .92 | 60.48 ± .11 | 46.63 ± .93 | 52.31 ± .69 | 51.72 ± 1.47 |

| (%) | LearnSPN | GeF | GeF(LSPN) | GeF$^+$ | GeF$^+$(LSPN) |
|---|---|---|---|---|---|
| 0 | 72.32 ± .20 | **91.22** ± .08 | **91.22** ± .08 | 90.46 ± .06 | 88.23 ± .13 |
| 10 | 71.14 ± .20 | 88.46 ± .08 | **88.76** ± .08 | 87.42 ± .06 | 84.75 ± .10 |
| 20 | 69.94 ± .15 | 85.45 ± .10 | **85.83** ± .10 | 84.31 ± .13 | 81.47 ± .11 |
| 30 | 68.61 ± .14 | 82.23 ± .12 | **82.64** ± .11 | 81.06 ± .14 | 78.39 ± .13 |
| 40 | 67.14 ± .14 | 78.88 ± .11 | **79.15** ± .10 | 77.83 ± .13 | 75.5 ± .16 |
| 50 | 65.7 ± .14 | **75.44** ± .15 | **75.55** ± .11 | 74.55 ± .13 | 72.78 ± .11 |
| 60 | 64.19 ± .16 | **71.94** ± .12 | **71.93** ± .10 | 71.25 ± .12 | 69.98 ± .14 |
| 70 | 62.54 ± .15 | **68.41** ± .14 | **68.33** ± .13 | 67.99 ± .12 | 67.12 ± .18 |
| 80 | 60.82 ± .11 | **64.92** ± .15 | **64.77** ± .15 | 64.65 ± .13 | 64.12 ± .23 |
| 90 | 59.13 ± .08 | **61.38** ± .12 | **61.25** ± .11 | **61.3** ± .12 | 61.01 ± .13 |

Figure 10: Accuracy against proportion of missing values. The same plot is repeated ten times, each time highlighting a different method and its 95% confidence interval.

## C.10 Gesture Phase Segmentation [13]

Dataset details

| n | $m_0$ | $m_1$ | $|\mathcal{Y}|$ | %Maj |
|---|---|---|---|---|
| 9873 | 0 | 32 | 5 | 29.88 |

Table 10: Accuracy per percent of missing values at test time with 95% confidence intervals.

| (%) | Surrogate | Friedman | Mean | KNN | MissForest |
|---|---|---|---|---|---|
| 0 | **65.9** ± .20 | **65.9** ± .20 | **65.9** ± .20 | **65.9** ± .20 | **65.9** ± .20 |
| 10 | 63.46 ± .16 | 60.7 ± .19 | 62.06 ± .21 | 64.69 ± .15 | **65.08** ± .19 |
| 20 | 61.08 ± .16 | 56.14 ± .20 | 58.6 ± .21 | 63.21 ± .13 | **63.64** ± .23 |
| 30 | 58.37 ± .15 | 52.86 ± .27 | 55.41 ± .21 | **61.62** ± .22 | 61.48 ± .26 |
| 40 | 55.79 ± .22 | 50.19 ± .17 | 52.44 ± .27 | **59.72** ± .33 | 58.89 ± .24 |
| 50 | 53.11 ± .35 | 48.16 ± .18 | 49.48 ± .38 | **57.46** ± .25 | 55.35 ± .33 |
| 60 | 50.67 ± .34 | 46.49 ± .14 | 46.52 ± .32 | **54.66** ± .26 | 51.18 ± .30 |
| 70 | 47.84 ± .20 | 44.93 ± .13 | 43.3 ± .29 | **51.37** ± .23 | 45.9 ± .27 |
| 80 | 44.69 ± .25 | 42.9 ± .18 | 39.33 ± .41 | **47.04** ± .22 | 39.36 ± .22 |
| 90 | 39.34 ± .60 | 38.99 ± .21 | 34.06 ± .39 | **40.85** ± .24 | 32.26 ± .34 |

| (%) | LearnSPN | GeF | GeF(LSPN) | GeF$^+$ | GeF$^+$(LSPN) |
|---|---|---|---|---|---|
| 0 | 42.15 ± .20 | **65.9** ± .20 | **65.9** ± .20 | 58.24 ± .40 | 55.69 ± .31 |
| 10 | 41.86 ± .19 | 63.78 ± .16 | **64.95** ± .18 | 56.67 ± .23 | 54.37 ± .14 |
| 20 | 41.53 ± .25 | 61.39 ± .16 | 62.9 ± .18 | 55.11 ± .25 | 53.23 ± .25 |
| 30 | 41.14 ± .24 | 58.65 ± .18 | 60.2 ± .21 | 53.5 ± .13 | 52.25 ± .20 |
| 40 | 40.78 ± .24 | 55.82 ± .16 | 57.42 ± .23 | 51.52 ± .21 | 50.97 ± .25 |
| 50 | 40.34 ± .25 | 52.92 ± .21 | 54.47 ± .28 | 49.64 ± .21 | 49.76 ± .23 |
| 60 | 39.78 ± .19 | 49.73 ± .21 | 51.37 ± .20 | 47.23 ± .30 | 47.97 ± .23 |
| 70 | 39.17 ± .27 | 46.66 ± .19 | 48.21 ± .19 | 44.75 ± .12 | 45.94 ± .18 |
| 80 | 38.24 ± .20 | 43.12 ± .21 | 44.79 ± .27 | 41.9 ± .19 | 43.32 ± .28 |
| 90 | 36.45 ± .17 | 39.42 ± .21 | **40.85** ± .22 | 38.76 ± .20 | 40.22 ± .18 |

Figure 11: Accuracy against proportion of missing values. The same plot is repeated ten times, each time highlighting a different method and its 95% confidence interval.

## C.11  Jungle Chess [21]

Dataset details

| n | $m_0$ | $m_1$ | $\|\mathcal{Y}\|$ | %Maj |
|---|---|---|---|---|
| 44819 | 6 | 0 | 3 | 51.46 |

Table 11: Accuracy per percent of missing values at test time with 95% confidence intervals.

| (%) | Surrogate | Friedman | Mean | KNN | MissForest |
|---|---|---|---|---|---|
| 0 | 85.66 ± .07 | 85.66 ± .07 | 85.66 ± .07 | 85.66 ± .07 | 85.66 ± .07 |
| 10 | 77.29 ± .10 | 80.4 ± .11 | 78.38 ± .13 | 77.98 ± .15 | 77.57 ± .11 |
| 20 | 69.9 ± .16 | 75.85 ± .13 | 72.11 ± .23 | 71.58 ± .18 | 71.05 ± .19 |
| 30 | 63.45 ± .24 | 71.91 ± .12 | 66.89 ± .40 | 66.25 ± .15 | 65.67 ± .20 |
| 40 | 57.82 ± .25 | 68.46 ± .13 | 62.5 ± .52 | 61.82 ± .17 | 61.29 ± .28 |
| 50 | 52.86 ± .27 | 65.36 ± .12 | 58.79 ± .67 | 57.96 ± .15 | 57.72 ± .36 |
| 60 | 48.49 ± .34 | 62.48 ± .10 | 55.7 ± .86 | 54.64 ± .35 | 54.52 ± .40 |
| 70 | 44.76 ± .32 | 59.68 ± .20 | 52.93 ± 1.02 | 51.79 ± .45 | 51.1 ± .67 |
| 80 | 41.85 ± .23 | 56.96 ± .24 | 50.66 ± 1.21 | 49.33 ± .63 | 48.13 ± 1.24 |
| 90 | 39.8 ± .12 | 54.22 ± .10 | 48.48 ± 1.46 | 47.49 ± 1.02 | 46.31 ± 1.54 |

| (%) | LearnSPN | GeF | GeF(LSPN) | GeF$^+$ | GeF$^+$(LSPN) |
|---|---|---|---|---|---|
| 0 | 77.03 ± .19 | 85.66 ± .07 | 85.64 ± .07 | **86.1** ± .07 | 85.97 ± .09 |
| 10 | 74.79 ± .16 | 80.35 ± .11 | 80.26 ± .10 | **80.53** ± .11 | 80.37 ± .11 |
| 20 | 72.42 ± .13 | **76.04** ± .14 | 75.92 ± .12 | **76.08** ± .14 | 75.92 ± .14 |
| 30 | 70.12 ± .14 | **72.4** ± .12 | **72.3** ± .11 | **72.39** ± .12 | 72.26 ± .12 |
| 40 | 67.79 ± .13 | **69.24** ± .15 | **69.17** ± .15 | **69.23** ± .15 | **69.15** ± .16 |
| 50 | 65.43 ± .14 | **66.3** ± .12 | **66.27** ± .13 | **66.3** ± .13 | **66.26** ± .13 |
| 60 | 63.0 ± .13 | **63.52** ± .12 | **63.5** ± .13 | **63.52** ± .12 | **63.49** ± .13 |
| 70 | 60.35 ± .19 | **60.59** ± .21 | **60.58** ± .22 | **60.59** ± .22 | **60.58** ± .22 |
| 80 | **57.58** ± .25 | **57.68** ± .26 | **57.68** ± .26 | **57.68** ± .26 | **57.68** ± .26 |
| 90 | **54.63** ± .11 | **54.67** ± .13 | **54.68** ± .13 | **54.67** ± .13 | **54.67** ± .13 |

Figure 12: Accuracy against proportion of missing values. The same plot is repeated ten times, each time highlighting a different method and its 95% confidence interval.

## C.12 King-Rook vs. King-Pawn (kr-vs-kp) [5]

Dataset details

| n | $m_0$ | $m_1$ | $|\mathcal{Y}|$ | %Maj |
|---|---|---|---|---|
| 3196 | 36 | 0 | 2 | 52.22 |

Table 12: Accuracy per percent of missing values at test time with 95% confidence intervals.

| (%) | Surrogate | Friedman | Mean | KNN | MissForest |
|---|---|---|---|---|---|
| 0 | **98.67** ± .10 | **98.67** ± .10 | **98.67** ± .10 | **98.67** ± .10 | **98.67** ± .10 |
| 10 | 89.24 ± .51 | 92.38 ± .26 | 93.67 ± .41 | **94.92** ± .42 | 94.58 ± .30 |
| 20 | 80.79 ± .96 | 87.35 ± .60 | 88.74 ± .63 | **91.08** ± .40 | 90.6 ± .34 |
| 30 | 73.62 ± .92 | 82.81 ± .73 | 83.58 ± .64 | **86.58** ± .43 | 86.24 ± .56 |
| 40 | 67.85 ± .89 | 78.8 ± .74 | 78.73 ± .71 | **82.31** ± .54 | 81.4 ± .31 |
| 50 | 63.39 ± 1.18 | 75.27 ± .75 | 74.17 ± .72 | 77.23 ± .52 | **75.95** ± .56 |
| 60 | 60.02 ± 1.09 | 71.45 ± .65 | 69.54 ± .53 | **72.27** ± .35 | 70.73 ± .47 |
| 70 | 57.38 ± .83 | **67.69** ± .60 | 65.12 ± .49 | 66.32 ± .56 | 65.15 ± .39 |
| 80 | 55.3 ± .80 | **63.44** ± .51 | 60.7 ± .42 | 60.39 ± .69 | 59.74 ± .72 |
| 90 | 53.36 ± .51 | **58.2** ± .58 | 56.31 ± .43 | 54.72 ± .72 | 54.81 ± .50 |

| (%) | LearnSPN | GeF | GeF(LSPN) | GeF$^+$ | GeF$^+$(LSPN) |
|---|---|---|---|---|---|
| 0 | 88.1 ± .67 | **98.67** ± .10 | **98.73** ± .09 | 98.08 ± .17 | 98.44 ± .12 |
| 10 | 86.03 ± .71 | **95.16** ± .28 | **95.19** ± .29 | 94.64 ± .33 | 94.88 ± .26 |
| 20 | 83.52 ± .68 | **92.06** ± .29 | **92.25** ± .30 | 91.54 ± .38 | 91.92 ± .30 |
| 30 | 80.92 ± .61 | **88.35** ± .39 | **88.65** ± .32 | 88.13 ± .38 | **88.41** ± .39 |
| 40 | 78.25 ± .44 | **84.72** ± .52 | **85.04** ± .45 | 84.49 ± .55 | **84.96** ± .49 |
| 50 | 75.46 ± .38 | **80.82** ± .30 | **81.15** ± .43 | 80.71 ± .25 | **81.05** ± .43 |
| 60 | 72.32 ± .46 | **76.17** ± .47 | **76.44** ± .43 | **76.2** ± .47 | **76.4** ± .43 |
| 70 | 68.78 ± .45 | **71.33** ± .45 | **71.54** ± .53 | **71.31** ± .46 | **71.46** ± .52 |
| 80 | 64.62 ± .32 | **65.92** ± .41 | **66.07** ± .51 | **65.88** ± .43 | **66.06** ± .49 |
| 90 | **59.56** ± .82 | **60.15** ± .74 | **60.17** ± .74 | **60.15** ± .73 | **60.17** ± .74 |

Figure 13: Accuracy against proportion of missing values. The same plot is repeated ten times, each time highlighting a different method and its 95% confidence interval.

## C.13   Mice Protein [9]

Dataset details

| n | $m_0$ | $m_1$ | $|\mathcal{Y}|$ | %Maj |
|---|---|---|---|---|
| 1080 | 0 | 77 | 8 | 13.89 |

Table 13: Accuracy per percent of missing values at test time with 95% confidence intervals.

| (%) | Surrogate | Friedman | Mean | KNN | MissForest |
|---|---|---|---|---|---|
| 0 | 98.78 ± .21 | 98.78 ± .21 | 98.78 ± .21 | 98.78 ± .21 | 98.78 ± .21 |
| 10 | 98.22 ± .18 | 97.11 ± .43 | 96.22 ± .33 | 98.61 ± .23 | 98.16 ± .24 |
| 20 | 97.31 ± .27 | 94.58 ± .38 | 92.18 ± .83 | 98.32 ± .31 | 97.29 ± .35 |
| 30 | 95.84 ± .44 | 91.01 ± .77 | 84.91 ± 1.03 | 97.7 ± .50 | 96.08 ± .52 |
| 40 | 93.53 ± .63 | 86.41 ± 1.09 | 74.03 ± 1.20 | 96.66 ± .49 | 93.96 ± .50 |
| 50 | 89.74 ± .75 | 80.3 ± .63 | 59.0 ± 1.79 | 94.69 ± .62 | 90.01 ± .75 |
| 60 | 83.82 ± .94 | 73.31 ± .77 | 44.34 ± 1.98 | 91.56 ± .58 | 82.94 ± .94 |
| 70 | 74.41 ± .91 | 63.41 ± .70 | 31.35 ± 1.39 | 86.21 ± .72 | 71.31 ± .97 |
| 80 | 60.07 ± .94 | 51.42 ± .76 | 21.81 ± 1.34 | **75.76** ± 1.30 | 53.63 ± 1.07 |
| 90 | 38.07 ± .98 | 34.98 ± 1.28 | 16.17 ± .78 | 54.88 ± .87 | 30.63 ± 1.46 |

| (%) | LearnSPN | GeF | GeF(LSPN) | GeF$^+$ | GeF$^+$(LSPN) |
|---|---|---|---|---|---|
| 0 | 93.82 ± .30 | 98.78 ± .21 | 98.78 ± .21 | **99.6** ± .19 | **99.65** ± .15 |
| 10 | 92.94 ± .31 | 98.93 ± .18 | 99.09 ± .17 | **99.31** ± .13 | **99.37** ± .12 |
| 20 | 92.01 ± .34 | 98.8 ± .10 | **99.26** ± .15 | 98.94 ± .24 | 99.03 ± .15 |
| 30 | 91.28 ± .35 | 98.38 ± .32 | **99.06** ± .15 | 98.05 ± .38 | 98.5 ± .34 |
| 40 | 89.8 ± .61 | 97.43 ± .37 | **98.78** ± .26 | 96.7 ± .34 | 97.8 ± .40 |
| 50 | 87.43 ± .55 | 95.15 ± .42 | **97.43** ± .33 | 94.41 ± .49 | 96.46 ± .44 |
| 60 | 85.09 ± .59 | 90.57 ± .65 | **94.34** ± .36 | 89.27 ± .57 | 93.13 ± .71 |
| 70 | 80.69 ± .69 | 81.66 ± .80 | **87.34** ± .77 | 80.4 ± 1.04 | 86.25 ± .73 |
| 80 | 73.6 ± .83 | 68.19 ± 1.11 | 73.06 ± 1.14 | 66.86 ± 1.18 | 71.96 ± 1.17 |
| 90 | **55.85** ± .72 | 46.83 ± .81 | 48.25 ± .94 | 46.24 ± .63 | 47.62 ± 1.07 |

Figure 14: Accuracy against proportion of missing values. The same plot is repeated ten times, each time highlighting a different method and its 95% confidence interval.

## C.14 Phishing Websites [5]

Dataset details

| n | $m_0$ | $m_1$ | $|\mathcal{Y}|$ | %Maj |
|---|---|---|---|---|
| 11055 | 30 | 0 | 2 | 55.69 |

Table 14: Accuracy per percent of missing values at test time with 95% confidence intervals.

| (%) | Surrogate | Friedman | Mean | KNN | MissForest |
|---|---|---|---|---|---|
| 0 | $96.74 \pm .05$ | $96.74 \pm .05$ | $96.74 \pm .05$ | $96.74 \pm .05$ | $96.74 \pm .05$ |
| 10 | $91.31 \pm .27$ | $93.88 \pm .11$ | $94.48 \pm .12$ | $95.48 \pm .07$ | $95.71 \pm .11$ |
| 20 | $86.08 \pm .41$ | $91.38 \pm .11$ | $91.68 \pm .12$ | $93.93 \pm .10$ | $93.91 \pm .12$ |
| 30 | $81.52 \pm .50$ | $88.98 \pm .17$ | $88.02 \pm .19$ | $92.06 \pm .13$ | $91.18 \pm .18$ |
| 40 | $77.57 \pm .56$ | $86.61 \pm .19$ | $83.8 \pm .20$ | $89.87 \pm .21$ | $87.57 \pm .24$ |
| 50 | $74.09 \pm .74$ | $83.91 \pm .28$ | $79.27 \pm .17$ | $86.94 \pm .23$ | $83.04 \pm .26$ |
| 60 | $70.73 \pm .73$ | $81.03 \pm .23$ | $74.22 \pm .31$ | $83.04 \pm .21$ | $77.79 \pm .36$ |
| 70 | $67.34 \pm .72$ | $77.11 \pm .26$ | $69.18 \pm .34$ | $78.2 \pm .36$ | $72.25 \pm .40$ |
| 80 | $63.28 \pm 1.01$ | $72.09 \pm .30$ | $64.28 \pm .42$ | $71.44 \pm .22$ | $65.98 \pm .34$ |
| 90 | $58.36 \pm 1.72$ | $64.72 \pm .32$ | $59.71 \pm .37$ | $62.35 \pm .25$ | $60.12 \pm .39$ |

| (%) | LearnSPN | GeF | GeF(LSPN) | GeF$^+$ | GeF$^+$(LSPN) |
|---|---|---|---|---|---|
| 0 | $93.33 \pm .15$ | $96.74 \pm .05$ | $96.78 \pm .06$ | $96.97 \pm .05$ | $\mathbf{97.06} \pm .06$ |
| 10 | $92.45 \pm .13$ | $95.85 \pm .12$ | $95.99 \pm .12$ | $96.17 \pm .07$ | $\underline{\mathbf{96.3}} \pm .09$ |
| 20 | $91.38 \pm .14$ | $94.67 \pm .09$ | $94.89 \pm .11$ | $95.04 \pm .10$ | $\underline{\mathbf{95.2}} \pm .11$ |
| 30 | $89.99 \pm .19$ | $92.99 \pm .08$ | $93.3 \pm .06$ | $93.41 \pm .11$ | $\underline{\mathbf{93.61}} \pm .10$ |
| 40 | $88.04 \pm .25$ | $90.88 \pm .16$ | $91.29 \pm .14$ | $91.22 \pm .17$ | $\underline{\mathbf{91.52}} \pm .18$ |
| 50 | $85.65 \pm .21$ | $87.98 \pm .19$ | $\mathbf{88.59} \pm .18$ | $88.32 \pm .17$ | $\underline{\mathbf{88.76}} \pm .18$ |
| 60 | $82.82 \pm .16$ | $84.58 \pm .15$ | $\mathbf{85.23} \pm .15$ | $84.79 \pm .13$ | $\underline{\mathbf{85.29}} \pm .16$ |
| 70 | $78.79 \pm .19$ | $80.1 \pm .21$ | $\mathbf{80.57} \pm .22$ | $80.21 \pm .19$ | $\underline{\mathbf{80.59}} \pm .20$ |
| 80 | $73.52 \pm .27$ | $\mathbf{74.29} \pm .35$ | $\mathbf{74.51} \pm .33$ | $\mathbf{74.31} \pm .32$ | $\underline{\mathbf{74.52}} \pm .32$ |
| 90 | $66.11 \pm .29$ | $\mathbf{66.39} \pm .40$ | $\underline{\mathbf{66.49}} \pm .34$ | $\mathbf{66.39} \pm .37$ | $\mathbf{66.48} \pm .33$ |

Figure 15: Accuracy against proportion of missing values. The same plot is repeated ten times, each time highlighting a different method and its 95% confidence interval.

## C.15 Robot (Wall-Following Robot Navigation) [5]

Dataset details

| n | $m_0$ | $m_1$ | $|\mathcal{Y}|$ | %Maj |
|---|---|---|---|---|
| 5456 | 0 | 24 | 4 | 40.41 |

Table 15: Accuracy per percent of missing values at test time with 95% confidence intervals.

| (%) | Surrogate | Friedman | Mean | KNN | MissForest |
|---|---|---|---|---|---|
| 0 | **99.46** ± .02 | **99.46** ± .02 | **99.46** ± .02 | **99.46** ± .02 | **99.46** ± .02 |
| 10 | 97.3 ± .15 | 95.58 ± .15 | 96.68 ± .14 | 97.38 ± .16 | 97.57 ± .11 |
| 20 | 94.77 ± .20 | 90.62 ± .21 | 93.4 ± .28 | 95.19 ± .11 | 95.07 ± .21 |
| 30 | 91.74 ± .23 | 84.73 ± .57 | 89.39 ± .28 | 92.74 ± .25 | 91.72 ± .37 |
| 40 | 87.98 ± .31 | 78.84 ± .62 | 84.55 ± .31 | 90.37 ± .30 | 87.07 ± .35 |
| 50 | 83.15 ± .32 | 73.14 ± .72 | 78.81 ± .45 | 87.79 ± .29 | 80.85 ± .45 |
| 60 | 77.11 ± .30 | 67.97 ± .70 | 72.48 ± .53 | **85.76** ± .20 | 72.51 ± .43 |
| 70 | 69.7 ± .33 | 63.27 ± .58 | 65.49 ± .40 | **83.63** ± .23 | 61.74 ± .51 |
| 80 | 60.84 ± .36 | 58.5 ± .65 | 57.95 ± .39 | **80.11** ± .33 | 47.96 ± .74 |
| 90 | 50.93 ± .26 | 52.55 ± .53 | 49.64 ± .29 | **68.29** ± .28 | 31.03 ± .88 |

| (%) | LearnSPN | GeF | GeF(LSPN) | GeF$^+$ | GeF$^+$(LSPN) |
|---|---|---|---|---|---|
| 0 | 79.41 ± .41 | **99.46** ± .02 | **99.47** ± .02 | 95.34 ± .17 | 94.75 ± .20 |
| 10 | 78.33 ± .36 | 97.57 ± .13 | **98.22** ± .10 | 94.31 ± .20 | 93.74 ± .20 |
| 20 | 76.69 ± .46 | 95.49 ± .19 | **96.7** ± .14 | 92.98 ± .19 | 92.71 ± .15 |
| 30 | 74.9 ± .37 | 92.97 ± .28 | **94.67** ± .20 | 90.87 ± .16 | 91.4 ± .12 |
| 40 | 72.31 ± .28 | 89.86 ± .20 | **91.97** ± .11 | 88.02 ± .17 | 89.58 ± .24 |
| 50 | 68.88 ± .29 | 85.72 ± .20 | **88.12** ± .26 | 83.93 ± .16 | 86.48 ± .20 |
| 60 | 64.75 ± .46 | 80.16 ± .27 | 82.78 ± .27 | 78.52 ± .23 | 81.8 ± .19 |
| 70 | 59.62 ± .43 | 72.79 ± .26 | 75.41 ± .41 | 71.32 ± .28 | 74.48 ± .29 |
| 80 | 54.29 ± .47 | 64.16 ± .37 | 65.86 ± .48 | 62.99 ± .28 | 65.06 ± .48 |
| 90 | 48.22 ± .49 | 53.73 ± .46 | 54.64 ± .26 | 53.28 ± .51 | 54.12 ± .30 |

Figure 16: Accuracy against proportion of missing values. The same plot is repeated ten times, each time highlighting a different method and its 95% confidence interval.

## C.16 Segment [5]

### Dataset details

| n | $m_0$ | $m_1$ | $|\mathcal{Y}|$ | %Maj |
|---|---|---|---|---|
| 2310 | 2 | 15 | 7 | 14.29 |

Table 16: Accuracy per percent of missing values at test time with 95% confidence intervals.

| (%) | Surrogate | Friedman | Mean | KNN | MissForest |
|---|---|---|---|---|---|
| 0 | **96.75** ± .20 | **96.75** ± .20 | **96.75** ± .20 | **96.75** ± .20 | **96.75** ± .20 |
| 10 | 95.86 ± .20 | 94.36 ± .29 | 92.51 ± .43 | **96.03** ± .20 | **96.15** ± .24 |
| 20 | 94.73 ± .30 | 90.02 ± .62 | 86.33 ± .77 | **95.09** ± .25 | **95.12** ± .26 |
| 30 | 93.32 ± .27 | 84.14 ± .69 | 78.34 ± .68 | **94.25** ± .32 | 93.21 ± .41 |
| 40 | 91.19 ± .41 | 77.55 ± .76 | 67.74 ± .76 | **93.03** ± .39 | 89.53 ± .57 |
| 50 | 88.05 ± .33 | 70.51 ± .64 | 55.79 ± 1.20 | **91.37** ± .55 | 83.65 ± .74 |
| 60 | 83.01 ± .52 | 62.86 ± .63 | 43.73 ± 1.18 | **88.48** ± .52 | 74.45 ± .86 |
| 70 | 74.73 ± .72 | 54.8 ± .43 | 32.73 ± .89 | **83.65** ± .79 | 62.03 ± 1.08 |
| 80 | 61.05 ± 1.10 | 45.88 ± .52 | 23.55 ± .72 | **73.19** ± .76 | 46.61 ± 1.46 |
| 90 | 40.65 ± .88 | 33.5 ± 1.13 | 17.44 ± .23 | **52.63** ± .66 | 30.55 ± 1.12 |

| (%) | LearnSPN | GeF | GeF(LSPN) | GeF$^+$ | GeF$^+$(LSPN) |
|---|---|---|---|---|---|
| 0 | 87.2 ± .12 | **96.75** ± .20 | **96.76** ± .20 | 94.12 ± .33 | 93.18 ± .33 |
| 10 | 85.98 ± .18 | 95.78 ± .20 | 95.88 ± .22 | 93.2 ± .37 | 90.84 ± .48 |
| 20 | 84.3 ± .28 | 94.68 ± .40 | 94.77 ± .38 | 91.93 ± .39 | 88.37 ± .50 |
| 30 | 82.58 ± .33 | 93.42 ± .20 | 93.41 ± .34 | 90.15 ± .56 | 86.26 ± .41 |
| 40 | 80.47 ± .34 | 91.19 ± .27 | 91.4 ± .34 | 88.0 ± .47 | 83.84 ± .64 |
| 50 | 77.48 ± .56 | 88.15 ± .33 | 88.26 ± .36 | 84.47 ± .54 | 81.0 ± .75 |
| 60 | 73.29 ± .59 | 83.45 ± .41 | 83.35 ± .46 | 79.68 ± .62 | 76.94 ± .67 |
| 70 | 67.0 ± .87 | 76.47 ± .87 | 76.1 ± .73 | 72.69 ± 1.03 | 71.1 ± .92 |
| 80 | 57.91 ± .68 | 64.91 ± 1.14 | 64.5 ± .92 | 62.32 ± 1.20 | 61.49 ± 1.06 |
| 90 | 42.73 ± .62 | 46.52 ± 1.10 | 45.98 ± .91 | 45.51 ± 1.07 | 45.07 ± .82 |

Figure 17: Accuracy against proportion of missing values. The same plot is repeated ten times, each time highlighting a different method and its 95% confidence interval.

## C.17 Splice (Primate splice-junction gene sequences) [5]

### Dataset details

| n | $m_0$ | $m_1$ | $|\mathcal{Y}|$ | %Maj |
|---|---|---|---|---|
| 3190 | 60 | 0 | 3 | 51.88 |

Table 17: Accuracy per percent of missing values at test time with 95% confidence intervals.

| (%) | Surrogate | Friedman | Mean | KNN | MissForest |
|---|---|---|---|---|---|
| 0 | **96.73** ± .12 | **96.73** ± .12 | **96.73** ± .12 | **96.73** ± .12 | **96.73** ± .12 |
| 10 | 94.48 ± .26 | 93.66 ± .32 | 94.55 ± .18 | 94.98 ± .18 | 94.8 ± .21 |
| 20 | 91.05 ± .36 | 89.79 ± .55 | 90.62 ± .26 | 92.35 ± .20 | 91.23 ± .30 |
| 30 | 86.09 ± .46 | 84.76 ± .65 | 84.65 ± .28 | 89.06 ± .53 | 86.18 ± .26 |
| 40 | 80.16 ± .36 | 79.12 ± .36 | 77.0 ± .43 | 84.8 ± .41 | 79.05 ± .48 |
| 50 | 73.44 ± .63 | 72.84 ± .40 | 67.98 ± .41 | 80.34 ± .40 | 70.4 ± .42 |
| 60 | 66.98 ± .95 | 66.56 ± .38 | 57.86 ± .51 | 74.04 ± .56 | 60.34 ± .48 |
| 70 | 61.12 ± .90 | 60.67 ± .59 | 47.52 ± .54 | 67.33 ± .48 | 49.54 ± .49 |
| 80 | 56.39 ± .64 | 56.03 ± .43 | 38.43 ± .30 | 59.43 ± .67 | 39.45 ± .33 |
| 90 | 53.12 ± .36 | 52.96 ± .14 | 30.23 ± .30 | 49.2 ± .46 | 30.57 ± .31 |

| (%) | LearnSPN | GeF | GeF(LSPN) | GeF$^+$ | GeF$^+$(LSPN) |
|---|---|---|---|---|---|
| 0 | 93.32 ± .22 | **96.73** ± .12 | **96.71** ± .12 | 96.6 ± .14 | 93.98 ± .29 |
| 10 | 92.42 ± .16 | 94.7 ± .32 | 93.27 ± .33 | **95.4** ± .17 | 92.64 ± .29 |
| 20 | 91.41 ± .34 | 92.88 ± .49 | 89.54 ± .43 | **93.98** ± .29 | 91.58 ± .28 |
| 30 | 89.8 ± .51 | 91.1 ± .50 | 85.69 ± .53 | **92.37** ± .32 | 90.38 ± .48 |
| 40 | 87.61 ± .40 | 88.93 ± .58 | 82.68 ± .33 | **90.41** ± .49 | 88.87 ± .43 |
| 50 | 85.05 ± .39 | 86.43 ± .50 | 80.71 ± .25 | **87.85** ± .40 | 86.72 ± .56 |
| 60 | 81.23 ± .46 | 83.18 ± .34 | 79.6 ± .27 | **84.33** ± .46 | 83.7 ± .43 |
| 70 | 76.05 ± .48 | 78.89 ± .37 | 77.52 ± .34 | **79.85** ± .40 | **79.88** ± .25 |
| 80 | 69.54 ± .58 | 72.32 ± .51 | 72.97 ± .53 | **73.78** ± .57 | **73.83** ± .50 |
| 90 | 61.03 ± .60 | 62.27 ± .69 | **63.01** ± .61 | **63.29** ± .71 | **63.72** ± .76 |

Figure 18: Accuracy against proportion of missing values. The same plot is repeated ten times, each time highlighting a different method and its 95% confidence interval.

## C.18 Texture [4]

Dataset details

| n | $m_0$ | $m_1$ | $|\mathcal{Y}|$ | %Maj |
|---|---|---|---|---|
| 5500 | 0 | 40 | 11 | 9.09 |

Table 18: Accuracy per percent of missing values at test time with 95% confidence intervals.

| (%) | Surrogate | Friedman | Mean | KNN | MissForest |
|---|---|---|---|---|---|
| 0 | $97.42 \pm .09$ | $97.42 \pm .09$ | $97.42 \pm .09$ | $97.42 \pm .09$ | $97.42 \pm .09$ |
| 10 | $96.98 \pm .10$ | $96.11 \pm .11$ | $95.64 \pm .18$ | $97.43 \pm .12$ | $97.14 \pm .11$ |
| 20 | $96.28 \pm .14$ | $93.79 \pm .24$ | $91.44 \pm .38$ | $97.32 \pm .15$ | $96.48 \pm .07$ |
| 30 | $95.31 \pm .15$ | $89.85 \pm .34$ | $84.24 \pm .42$ | $\mathbf{97.13} \pm .15$ | $95.4 \pm .17$ |
| 40 | $93.98 \pm .12$ | $84.52 \pm .42$ | $74.08 \pm .54$ | $\mathbf{96.83} \pm .10$ | $93.23 \pm .20$ |
| 50 | $91.82 \pm .14$ | $77.7 \pm .44$ | $61.24 \pm .77$ | $\mathbf{96.48} \pm .15$ | $89.48 \pm .40$ |
| 60 | $88.53 \pm .22$ | $69.65 \pm .69$ | $47.14 \pm 1.04$ | $\mathbf{95.76} \pm .14$ | $82.83 \pm .71$ |
| 70 | $82.42 \pm .47$ | $61.2 \pm .57$ | $32.83 \pm .78$ | $\mathbf{94.21} \pm .08$ | $70.86 \pm .59$ |
| 80 | $70.0 \pm .67$ | $51.64 \pm .63$ | $20.39 \pm .89$ | $\mathbf{89.58} \pm .31$ | $52.67 \pm .68$ |
| 90 | $44.8 \pm .93$ | $39.0 \pm .72$ | $12.33 \pm .49$ | $\mathbf{71.34} \pm .56$ | $29.33 \pm .76$ |

| (%) | LearnSPN | GeF | GeF(LSPN) | GeF$^+$ | GeF$^+$(LSPN) |
|---|---|---|---|---|---|
| 0 | $76.36 \pm .09$ | $97.42 \pm .09$ | $97.43 \pm .08$ | $96.43 \pm .10$ | $\mathbf{97.65} \pm .09$ |
| 10 | $75.75 \pm .12$ | $97.18 \pm .08$ | $\mathbf{97.58} \pm .11$ | $95.79 \pm .08$ | $97.32 \pm .10$ |
| 20 | $74.89 \pm .15$ | $96.64 \pm .09$ | $\mathbf{97.47} \pm .08$ | $95.08 \pm .11$ | $97.11 \pm .15$ |
| 30 | $73.98 \pm .19$ | $95.93 \pm .15$ | $\mathbf{97.12} \pm .13$ | $94.2 \pm .10$ | $96.67 \pm .13$ |
| 40 | $73.06 \pm .23$ | $94.75 \pm .17$ | $96.44 \pm .11$ | $92.87 \pm .11$ | $95.97 \pm .18$ |
| 50 | $71.94 \pm .24$ | $92.92 \pm .22$ | $95.2 \pm .15$ | $91.02 \pm .18$ | $94.81 \pm .13$ |
| 60 | $70.48 \pm .26$ | $90.34 \pm .25$ | $93.07 \pm .21$ | $88.62 \pm .27$ | $92.69 \pm .26$ |
| 70 | $68.38 \pm .32$ | $86.16 \pm .29$ | $88.89 \pm .24$ | $84.84 \pm .26$ | $88.48 \pm .27$ |
| 80 | $64.88 \pm .46$ | $78.98 \pm .42$ | $80.9 \pm .41$ | $78.01 \pm .39$ | $80.51 \pm .41$ |
| 90 | $55.35 \pm .40$ | $62.95 \pm .47$ | $63.57 \pm .37$ | $62.45 \pm .48$ | $63.37 \pm .39$ |

Figure 19: Accuracy against proportion of missing values. The same plot is repeated ten times, each time highlighting a different method and its 95% confidence interval.

[4]This database was generated by the Laboratory of Image Processing and Pattern Recognition (INPG-LTIRF) in the development of the Esprit project ELENA No. 6891 and the Esprit working group ATHOS No. 6620.

## C.19 Vehicle [18]

Dataset details

| n | $m_0$ | $m_1$ | $|\mathcal{Y}|$ | %Maj |
|---|---|---|---|---|
| 846 | 0 | 18 | 4 | 25.77 |

Table 19: Accuracy per percent of missing values at test time with 95% confidence intervals.

| (%) | Surrogate | Friedman | Mean | KNN | MissForest |
|---|---|---|---|---|---|
| 0 | **75.06** ± .80 | **75.06** ± .80 | **75.06** ± .80 | **75.06** ± .80 | **75.06** ± .80 |
| 10 | **74.37** ± .58 | 73.18 ± .69 | 72.27 ± .99 | **74.21** ± 1.00 | **74.19** ± .71 |
| 20 | **72.98** ± .84 | 70.1 ± .70 | 68.06 ± 1.12 | **72.84** ± .86 | **72.57** ± .90 |
| 30 | **71.61** ± .92 | 67.12 ± .79 | 63.27 ± 1.05 | **71.77** ± 1.01 | 70.69 ± 1.33 |
| 40 | 69.96 ± 1.14 | 63.65 ± .82 | 57.53 ± .88 | 70.16 ± 1.16 | 68.72 ± 1.39 |
| 50 | 67.18 ± 1.47 | 59.98 ± 1.28 | 52.06 ± 1.42 | **68.31** ± 1.03 | 64.9 ± 1.36 |
| 60 | 63.91 ± 1.27 | 56.08 ± 1.11 | 45.67 ± 1.53 | **66.17** ± 1.04 | 59.76 ± 1.36 |
| 70 | 58.65 ± 1.06 | 51.13 ± 1.03 | 38.74 ± 1.63 | **62.33** ± 1.31 | 53.48 ± 1.57 |
| 80 | 50.89 ± 1.39 | 46.54 ± .96 | 33.34 ± 1.58 | **56.08** ± 1.15 | 45.02 ± 1.03 |
| 90 | 38.69 ± .90 | 38.74 ± 1.16 | 28.51 ± .65 | **44.9** ± 1.04 | 36.26 ± .96 |

| (%) | LearnSPN | GeF | GeF(LSPN) | GeF$^+$ | GeF$^+$(LSPN) |
|---|---|---|---|---|---|
| 0 | 65.98 ± .96 | **75.06** ± .80 | **75.07** ± .80 | 73.12 ± .73 | 73.26 ± .74 |
| 10 | 64.7 ± 1.03 | **73.98** ± .74 | **74.23** ± .88 | 72.15 ± .61 | 72.47 ± .88 |
| 20 | 63.67 ± 1.05 | **73.1** ± .88 | **73.41** ± .97 | 71.42 ± .75 | 71.9 ± .76 |
| 30 | 62.39 ± .99 | **72.39** ± 1.13 | **72.77** ± 1.27 | 70.56 ± 1.03 | 71.06 ± 1.15 |
| 40 | 61.04 ± 1.29 | **70.94** ± 1.21 | **71.52** ± 1.23 | 69.53 ± 1.03 | 69.78 ± 1.23 |
| 50 | 58.92 ± 1.44 | **68.6** ± 1.32 | **69.4** ± 1.34 | 67.05 ± 1.39 | 67.46 ± 1.47 |
| 60 | 56.86 ± 1.02 | **65.7** ± 1.10 | **66.0** ± .96 | 63.93 ± 1.22 | 64.16 ± 1.02 |
| 70 | 53.54 ± 1.20 | **61.17** ± .94 | **61.79** ± .95 | 59.67 ± .79 | 60.25 ± .92 |
| 80 | 48.25 ± .62 | **55.24** ± .72 | **55.09** ± 1.23 | 54.01 ± .66 | 54.14 ± 1.09 |
| 90 | 40.18 ± 1.39 | **44.17** ± .99 | **44.27** ± .71 | **44.01** ± .93 | 43.62 ± .99 |

Figure 20: Accuracy against proportion of missing values. The same plot is repeated ten times, each time highlighting a different method and its 95% confidence interval.

## C.20  Vowel [5]

Dataset details

| n | $m_0$ | $m_1$ | $|\mathcal{Y}|$ | %Maj |
|---|---|---|---|---|
| 990 | 2 | 10 | 11 | 9.09 |

Table 20: Accuracy per percent of missing values at test time with 95% confidence intervals.

| (%) | Surrogate | Friedman | Mean | KNN | MissForest |
|---|---|---|---|---|---|
| 0 | $96.07 \pm .51$ | $96.07 \pm .51$ | $96.07 \pm .51$ | $96.07 \pm .51$ | $96.07 \pm .51$ |
| 10 | $92.37 \pm .63$ | $88.99 \pm .80$ | $86.71 \pm .87$ | $93.31 \pm .87$ | $93.2 \pm .64$ |
| 20 | $86.26 \pm .50$ | $80.21 \pm 1.07$ | $75.83 \pm .56$ | $90.17 \pm .68$ | $88.3 \pm .71$ |
| 30 | $78.79 \pm .86$ | $70.81 \pm 1.45$ | $64.51 \pm .83$ | $85.62 \pm .66$ | $81.85 \pm 1.10$ |
| 40 | $68.82 \pm 1.33$ | $60.67 \pm 1.46$ | $51.97 \pm 1.53$ | $79.0 \pm .81$ | $73.43 \pm 1.47$ |
| 50 | $58.06 \pm 1.95$ | $51.41 \pm 1.10$ | $41.52 \pm 1.29$ | $70.68 \pm 1.01$ | $62.62 \pm 1.39$ |
| 60 | $47.05 \pm 1.63$ | $42.35 \pm 1.45$ | $32.44 \pm .87$ | $59.78 \pm 1.14$ | $51.19 \pm .80$ |
| 70 | $35.09 \pm 1.14$ | $33.34 \pm .72$ | $24.02 \pm .49$ | $46.19 \pm 1.19$ | $38.22 \pm .82$ |
| 80 | $25.15 \pm 1.49$ | $25.57 \pm .75$ | $17.55 \pm .63$ | $32.52 \pm 1.30$ | $26.24 \pm 1.37$ |
| 90 | $16.55 \pm .90$ | $18.58 \pm .95$ | $12.71 \pm .62$ | $18.61 \pm .93$ | $16.41 \pm .89$ |

| (%) | LearnSPN | GeF | GeF(LSPN) | GeF$^+$ | GeF$^+$(LSPN) |
|---|---|---|---|---|---|
| 0 | $60.01 \pm 1.00$ | $96.07 \pm .51$ | $96.07 \pm .51$ | $\mathbf{97.38} \pm .36$ | $\mathbf{97.33} \pm .34$ |
| 10 | $56.64 \pm 1.19$ | $95.09 \pm .44$ | $95.15 \pm .42$ | $\mathbf{95.75} \pm .41$ | $\mathbf{95.85} \pm .41$ |
| 20 | $53.01 \pm 1.33$ | $92.73 \pm .48$ | $\mathbf{92.97} \pm .43$ | $\underline{\mathbf{93.39}} \pm .51$ | $\mathbf{93.39} \pm .50$ |
| 30 | $49.62 \pm 1.09$ | $\mathbf{89.25} \pm .77$ | $89.59 \pm .91$ | $89.46 \pm .69$ | $\mathbf{89.79} \pm .71$ |
| 40 | $44.04 \pm 1.17$ | $\mathbf{82.95} \pm .65$ | $\underline{\mathbf{83.48}} \pm .70$ | $82.86 \pm .63$ | $83.3 \pm .55$ |
| 50 | $39.48 \pm 1.42$ | $\mathbf{73.9} \pm .58$ | $\underline{\mathbf{74.52}} \pm .64$ | $73.27 \pm .53$ | $73.97 \pm .70$ |
| 60 | $34.94 \pm 1.35$ | $\mathbf{62.6} \pm .88$ | $\underline{\mathbf{63.16}} \pm .77$ | $61.93 \pm .68$ | $62.53 \pm .63$ |
| 70 | $29.06 \pm 1.14$ | $47.97 \pm .58$ | $\underline{\mathbf{48.52}} \pm .52$ | $47.21 \pm .76$ | $47.76 \pm .68$ |
| 80 | $23.09 \pm 1.35$ | $\mathbf{33.47} \pm 1.13$ | $\underline{\mathbf{33.71}} \pm 1.07$ | $\mathbf{33.05} \pm .99$ | $\mathbf{33.09} \pm .97$ |
| 90 | $17.46 \pm 1.03$ | $\underline{\mathbf{20.68}} \pm 1.23$ | $20.59 \pm 1.21$ | $20.28 \pm 1.20$ | $\mathbf{20.4} \pm 1.26$ |

Figure 21: Accuracy against proportion of missing values. The same plot is repeated ten times, each time highlighting a different method and its 95% confidence interval.

## C.21 Wine Quality [14]

Dataset details

| n | $m_0$ | $m_1$ | $|\mathcal{Y}|$ | %Maj |
|---|---|---|---|---|
| 6497 | 0 | 11 | 2 | 80.34 |

This dataset includes both red and white wine data. For classification purposes, the target variable was split into two classes: scores less or equal to 6, and scores greater than 6.

Table 21: Accuracy per percent of missing values at test time with 95% confidence intervals.

| (%) | Surrogate | Friedman | Mean | KNN | MissForest |
|---|---|---|---|---|---|
| 0 | **88.8** ± .13 | **88.8** ± .13 | **88.8** ± .13 | **88.8** ± .13 | **88.8** ± .13 |
| 10 | 87.41 ± .21 | 86.1 ± .15 | 86.61 ± .20 | 87.96 ± .16 | **88.21** ± .19 |
| 20 | 85.86 ± .18 | 83.96 ± .17 | 84.71 ± .16 | 86.93 ± .24 | **87.18** ± .22 |
| 30 | 84.49 ± .17 | 82.45 ± .10 | 83.2 ± .15 | 85.73 ± .26 | **85.95** ± .12 |
| 40 | 83.19 ± .15 | 81.39 ± .10 | 82.08 ± .09 | **84.5** ± .32 | 84.4 ± .21 |
| 50 | 82.16 ± .15 | 80.93 ± .11 | 81.32 ± .13 | **83.32** ± .23 | 82.98 ± .19 |
| 60 | 81.42 ± .20 | 80.62 ± .08 | 80.77 ± .13 | **82.24** ± .20 | 81.64 ± .16 |
| 70 | 80.86 ± .13 | 80.47 ± .04 | 80.47 ± .11 | **81.15** ± .23 | 80.58 ± .25 |
| 80 | 80.6 ± .12 | 80.38 ± .02 | 80.4 ± .13 | 80.29 ± .19 | 79.92 ± .24 |
| 90 | **80.42** ± .05 | 80.35 ± .01 | 80.35 ± .06 | 79.97 ± .18 | 79.77 ± .22 |

| (%) | LearnSPN | GeF | GeF(LSPN) | GeF$^+$ | GeF$^+$(LSPN) |
|---|---|---|---|---|---|
| 0 | 81.63 ± .14 | **88.8** ± .13 | **88.81** ± .13 | 87.83 ± .22 | 86.34 ± .18 |
| 10 | 81.44 ± .15 | 87.89 ± .16 | **88.18** ± .17 | 87.5 ± .25 | 86.55 ± .22 |
| 20 | 81.3 ± .16 | 86.62 ± .19 | **87.13** ± .22 | 86.8 ± .24 | 86.41 ± .25 |
| 30 | 81.08 ± .20 | 85.22 ± .18 | **85.85** ± .19 | 85.58 ± .18 | 85.69 ± .22 |
| 40 | 80.99 ± .13 | 83.77 ± .15 | 84.31 ± .18 | 84.19 ± .13 | **84.63** ± .15 |
| 50 | 80.78 ± .14 | 82.66 ± .16 | 82.98 ± .20 | 82.99 ± .17 | **83.29** ± .19 |
| 60 | 80.62 ± .15 | 81.75 ± .22 | 81.92 ± .18 | 81.87 ± .22 | **82.06** ± .22 |
| 70 | 80.52 ± .07 | **81.07** ± .14 | **81.13** ± .14 | **81.12** ± .16 | **81.23** ± .17 |
| 80 | 80.43 ± .02 | **80.66** ± .09 | **80.69** ± .09 | 80.67 ± .08 | **80.7** ± .09 |
| 90 | 80.38 ± .04 | **80.47** ± .07 | **80.47** ± .06 | **80.47** ± .07 | 80.46 ± .07 |

Figure 22: Accuracy against proportion of missing values. The same plot is repeated ten times, each time highlighting a different method and its 95% confidence interval.

## Footnotes

[1] Here, we assume for simplicity that all variables are continuous. Including discrete variables with finitely many states can be done by applying similar arguments to each of the finitely many joint states.

[2] https://www.openml.org/s/99/data