[Reviews · NeurIPS 2020]

Review 1

Summary and Contributions: This paper proposes a generative version of Decision Tree and Random Forests called GenDT i.e Generative Decision Trees and GenRF i.e Generate Random Forests. Decision Trees which are highly effective discriminative models are converted to probabilistic circuits to derive their generative counterparts. These generative counterparts are really effective for scenarios like in the presence of missing values, where values need to be filled explicitly before DTs are applied. This paper shows that the learned generative model is more effective than firstly filling missing values and running traditional DTs over those illustrating the effectiveness of the proposed model.

Strengths: This work seems an interesting connection between largely separated communities of Random Forests and Probabilistic circuits and seems to bridge the gap between two communities. The proposed solution seems to be simple, elegant and yet effective and looks to bridge the gap of taking DTs and RFs to generative models regime. The proposed method is effective in the presence of missing values and the proposed approach outperforms the traditional work of first filling in missing values and then running discriminative models over the top in most of the datasets I think this work is highly relevant to the practical ML community since DT and RF are models of choice where image, NLP or speech data is not present and some features of the model are still used and this type of work could be of immense value to the community.

Weaknesses: One weakness is that it may be hard to learn that many number of densities in leaves. I am wondering if there are some ideas which can exploit the shared structure among different leaves so that these densities can be learnt jointly. Also, this density estimation can become much more complex as the number of variables and labels scale. It would be great to have some discussion around this aspect. The work definitely casts an additional learning overhead in Decision trees of not only learning the splits but also densities. I would like to see some quantification of this additional computational overhead. I also wonder if the work may characterize when the densities should be modeled as SPNs or fully factorized leaves. Why in some scenarios, fully factorized leaves perform better than GF-SPNs and in those case, SPNs are not able to learn full factorization.

Correctness: Yes, the claims , method and empirical evaluations seem correct

Clarity: The paper is mostly well written and all important parts are clearly explained. It was an easy to follow paper with motivation and use case of the approach well explained. I am confused in the imputation example. Shouldn’t prediction goes to 0 as X 2 approaches E(X2/X1) instead of 1 as stated in the text. Is there some typo in this or another expression or a misunderstanding on my part. Figure 2 is slightly puzzling to understand at first glance. I presume it is showing same data but confidence intervals of the model in focus. I suggest to have only 1 figure instead of 4 figures as illustrated and that too with confidence intervals in that.

Relation to Prior Work: Yes, the related work section seems good enough with key differences clearly highlighted and explained wrt both decision tree literature and the connections to the probabilistic circuits community.

Reproducibility: Yes

Additional Feedback: Post Rebuttal: I read the author feedback and other reviews. Thanks for the feedback. I totally agree that the main contribution of the paper is making the two communities of SPN and DTs closer but R5 has still valid concerns regarding more experiments on hyper-parameter search and doing experiments to illustrate competitive advantage or where the method stands compared to existing work. I will suggest the authors to not take this in light of improving accuracy but to understand and make others understand the proposed work more clearly from all aspects. It will go to make the work more stronger and may have impact on long term usage of such methods in the community. So, i will strongly suggest authors to perform additional experiments to improve the work further.


Review 2

Summary and Contributions: The paper provides an algorithm to convert decision trees (DTs) to probabilistic circuits (PCs), with extensions to ensembles of DTs like random forests (RFs). This unites the discriminative and generative viewpoints. The most prominent advantage of the framework is the ability to deal with missing values at test time in a principled way. Experiments show that this method of handling missing values outperforms competitors (e.g., surrogate splits, imputation).

Strengths: As the paper points out, RFs are a widely used machine learning method, popular for their usability and flexibility. This paper contributes a way to incorporate the advantages of generative models into RFs so that they may handle missing values better. The algorithm seems to be simple and efficient, preserving the advantages of standard RFs.

Weaknesses: My only question was about whether there might be a way to compare more directly against the most related approaches in the related work section (e.g., DT+KDE, DT+NBC, CNets). However, the discussion in that section indicates that the methods have some fundamental differences. Perhaps the authors could say a bit more in the author response phase?

Correctness: The methodology seems sound.

Clarity: The writing is excellent.

Relation to Prior Work: The related work section seems good. See a question above about a bit more detail and/or direct comparison.

Reproducibility: Yes

Additional Feedback: Update post-feedback: I have read the other reviews and author feedback. I agree that there is room for improvement and that the authors should seriously consider the suggestions, but I think there is an interesting contribution here even if there is room for future work.


Review 3

Summary and Contributions: The paper proposes to add density models at decision tree leaves allowing them to make prediction of full joint densities. The paper shows that the resulting models, called generative decision trees (GeDT) and forests (GeRF), can be used to handle missing values at test time in an elegant way and also to detect outliers.

Strengths: 1. The idea of extending DT and RF so that they become fully generative model is very appealing. 2. The proposed approach is very simple and sound. It does not modify the standard DT and RF algorithms and its target predictions remain unchanged with respect to these two algorithms under some assumption. 3. Experiments show that the approach is very competitive against some existing methods for dealing with missing values at prediction time. Results are analysed using CV and appropriate statistical tests. 4. The authors provide a proof that their estimators are consistent whatever the pattern of missing values under the assumption of class factorization. This is a nice result.

Weaknesses: 1. While the approach is presented as a general generative model based on DT and RF, the paper fails to show its practical interest beyond handling missing values at test time. The possibility of using the approach for outlier detection is potentially interesting but the experiment in the paper is restricted to a single dataset and does not include any comparison with competitors except Gaussian KDE. Overall, the properties of GeDT and GeRF as general purpose density estimators are not really studied. My feeling is that because the tree partitioning is unchanged with respect to standard discriminative DT and RF, GeDT and GeRF are probably only appropriate in the context of tasks related to target predictions. In other tasks, I don't see why they would perform better than pure PC models or other methods mentioned in the related work section. In this sense, I think the paper oversells a bit the generality of GeDT and GeRF as full density models. 2. Empirical results are good with respect to the compared techniques. There are some limitations in these experiments however. First, an analysis of the impact of the RF hyper-parameters is missing, in particular parameters that restrict tree complexity (minimum number of samples in a leaf or maximum tree depth for example) and the number of randomly selected features used to split a node (RF's mtry). In the extreme case where trees are restricted to a single node, the approach is indeed equivalent to the density model used on the full dataset (plus bagging), while when the trees are fully grown, it becomes less dependent on this density model. Mtry also influences the dependence of the tree structure on the target Y and I think it should have an impact on the robustness against missing inputs (if mtry is high, there is less variability in the features that are selected in the trees, and therefore maybe less robustness with respect to missing values of the unselected inputs). GeDT are also not evaluated at all, only GeRF. I think a comparison with LearnSPN applied on the full dataset is also missing, to assess the benefit of first partitioning the space discriminatively using trees. 3. The paper gives a lot of details about the decision tree part and jts transformation but does not really explain enough the density model part (But I'm more familiar with trees and forests than with PC density models). See the clarity section below for some questions. 4. There are some missing related works. See below. Section 5 also mentions a number of decision tree based density estimators, such as density estimation trees or cutset networks that are close to GeF. The authors should better motivate why GeFs are not compared against these methods in the experimental section.

Correctness: Both theoretical and empirical results look correct to me.

Clarity: The paper is mostly clear but some details about the methods and the experiments are missing in the main text and are not really clarified in the supplementary material. - In the supplementary material, it is said about RF that "the only stop criterion is the impurity of the class variable, possibly leading to many leaves with a single sample". Does it mean that you stop splitting only when the impurity is zero or do you use a threshold on impurity? If you stop splitting a node only when impurity is zero, then the fact that LearnSPN is not class-factorized should not matter much as the output will be expected to be constant in almost all leaves. As already mentioned, I think it would be worth testing different pruning levels. - How is RF's mtry parameter set? (the number of features randomly selected at each node). - The term "fully factorized leaves" should be explained in the main text (unless I missed it, it's only explained in the supplementary material). - How are the distributions P(X_i) modelled in the case of fully factorized leaves? - I think that Friedman's method should be explained in the main paper and its difference with respect to GeF with fully factorized leaves should be better highlighted, as both methods appear to be quite close to me. Isn't Friedman's method strictly equivalent to fully factorized GeF when using uniform probabilities P(X_i)? I'm wondering also if Theorem 2 does not also apply to Friedman's method. I don't see why it wouldn't.

Relation to Prior Work: Some more sophisticated RF based missing value imputation methods are missing in the discussions. There is for example the missForest method: Steckhoven and Bühlmann, MissForest -- non-parametric missing value imputation for mixed-type data. Bioinformatics, 28(1):112-118, 2011. It's however more related to kNN imputation than to the proposed method. More related is MERCS proposed in : Van Wolputte et al. MERCS: multi-directional ensembles of regression and classification trees, AAAI 2018. Like GeRF, this method allows to predict a given output Y from any set of inputs and it has been exploited for handling missing value and outlier detection. One difference however is that MERCS does not require a priori to identify a specific output Y and can be used to predict any variable. I think this is potentially an important advantage over GeRF that needs to choose a specific output a priori. For outlier detection, a popular method is isolation forests: Liu, Fei Tony; Ting, Kai Ming; Zhou, Zhi-Hua, "Isolation Forest". 2008 Eighth IEEE International Conference on Data Mining: 413–422.

Reproducibility: Yes

Additional Feedback: Minor comments: - Line 132: "a fraction p of all variables […] p=0.3 o p=\sqrt{m}" \sqrt{m} is not a fraction but an absolute number. - Line 230 (and at other places): I'm not sure to understand why the authors need to make the assumption that the inputs are missing at random. This is not relevant given that the algorithm treats missing data at prediction time. ** Update after the authors' response ** I thank the authors to have responded to my main comments. I still believe that the idea is interesting and that the paper can be accepted, but my initial concerns remain. Below, some more feedback about some points in the authors' response. - "As our goal is not to find the most accurate RF/GeF, we use literature defaults and defer tuning to future work" My suggestion of studying the influence of the hyper-parameters is not to get the best possible model but to better understand the method. Figure 1 in the authors' response is one step in this direction but I think more experiments are still needed, for example changing RF's mtry parameters and looking at the impact of pruning on separate datasets (do you observe a monotonic decrease of accuracy on all problems?). I actually don't really understand why no pruning is the best setting, as it leaves very few samples in each leaf for fitting the - "P(Xi) is a marginal (multinomial/normal) trained on data matching the leaf. " "Note that Friedman’s method ignores the actual values of explanatory variables…" I would have liked to have a more explicit answer about the difference between Friedman's method and GeF with fully marginalized leaves and uniform marginals. Aren't they strictly similar? - "We agree that claiming GeFs are the new SOTA for treating missing values would require further experimentation, but we do not make such a claim." You indeed did not claim that GeRFs was the new SOTA for treating missing values but, again, since this is the only application really seriously explored in the paper, I would still have liked to see experiments against SOTA in this domain, especially based on trees/forests. Also, I believe that GeFs and MERCs addressed similar goals and it makes very much sense to compare them.

[Author Response · NeurIPS 2020]

We want to thank all reviewers for their time and effort. We address the raised points below.

- **R1** *hard to learn large number of densities*: We used a threshold on the number of samples at each leaf, to decide whether LearnSPN or a fully factorised density is used. This drastically reduces the computational effort to learn the leaves. In general, GeDTs/GeFs allow for much freedom, as any density estimator can be used at the leaves.

- **R1** *additional computational overhead*: The asymptotic costs of learning and inference are comparable to standard RFs (see supp. material). In general, the overhead depends on the particular choice of leaf estimators.

- **R1** *when to use SPNs or fully factorised leaves*: A key consideration is overfitting, and using factorised leaves induces a conditional (on the DT decisions) independence assumption which is more robust for few samples.

- **R1** *prediction goes to 0 as X 2 approaches E(X2/X1) instead of 1*: This is indeed a typo: Prediction will be (w.h.p.) $Y = 0$ (the argumentation remains valid as presented in the paper). Thanks for the catch!

- **R1** *Fig.2 is slightly puzzling*: We see the point, and will revise the figure.

- **R3** *compare against the most related approaches (e.g., DT+KDE, DT+NBC, CNets)*: Note that GeDTs/GeFs leaves can be equipped with any density model. Therefore, when we equip them with KDEs or NBCs, they simply subsume DT+KDE and DT+NBC in terms of classification. The crucial difference in our paper is, that these previous approaches did not follow through with the interpretation of the **whole** DT as a joint distribution and its advantages (missing data, outlier detection). We will include a comparison to CNets.

- **R5** *practical interest beyond handling missing values, properties of GeDT/GeRF*: We focused on classification under missing data since it is the most compelling advantage: DT/RF practitioners do not need to change the structure learning algorithm, but can now treat missing inputs within the same model, and with guaranteed backwards compatibility (provided $p(Y, \mathbf{X}) = p(Y)p(\mathbf{X})$ in all leaves). We will include more thorough experiments on outlier detection in the revised version. Our main contribution is that we link two separate communities, PCs and DTs, which will likely lead to ample cross-fertilisation of ideas.

- **R5** *RF hyper-parameters*: Our experiments only compare models with the same underlying structure, so variations in the RF hyper-parameters would most likely only change the basis for comparison but not the overall results. As our goal is not to find the most accurate RF/GeF, we use literature defaults and defer tuning to future work (non-tuned results are already very good). That said, we did experiment with a few hyper-parameters. For instance, pruning has not proved very useful (not uncommon in RFs), see right side of Figure below.

- **R5** *LearnSPN on the full dataset*: We experimented with a SOTA implementation of LearnSPN and results are quite poor. For the sake of space we omitted it in the submission (see left side of Figure below), but will include it given acceptance.

- **R5** *explanations on density model part*: GeDTs/GeFs can be used with any density estimator, which is subject to exploration in the future (we use SPNs as cited). Also, we will explain fully factorised leaves in the paper. $P(X_i)$ is a marginal (multinomial/normal) trained on data matching the leaf.

- **R5** *Friedman method*: We will clarify differences in the paper; results do not automatically apply, but it may be possible to extend them. Note that Friedman's method ignores the actual values of explanatory variables, while fully factorised leaves do compute $p(\mathbf{x})$ to handle missing values.

- **R5** *MissForest, MERCS, Isolation Forest*: We agree that claiming GeFs are the new SOTA for treating missing values would require further experimentation, but we do not make such a claim. In our experiments, we focus on 'built-in' methods as they are closer to ours in nature, and also compared to KNN because it is the most common imputation method and the closest to ours in computational cost. We will cite and discuss the works pointed out.

Figure 1: Average accuracy gain relative to RF+KNN imputation (left) and average accuracy on test data of GeF models with different minimum number of samples per leaf (right). Both averages are across the datasets considered.

[Meta-Review · NeurIPS 2020]

Overall, reviewers found the contribution significantly novel: the authors connect two disjoint domains (decision trees and probabilistic circuits), and demonstrated effectiveness of their approach on datasets with missing values. Two main concerns remain, even after the rebuttal (i) it's unclear how the proposed approach has advantages over existing alternatives (ii) the effect of the hyper-parameters remain unclear. Consensus after the discussion period was to accept.